# AR MODELS CAN BE FASTER AND MORE ACCURATE PARALLEL DECODERS THAN DIFFUSION LLMS

## ABSTRACT

Multi-token generation has emerged as a promising paradigm for accelerating transformer-based large model inference. Recent efforts have primarily explored diffusion-based LLMs (dLLM) for parallel decoding to reduce latency while preserving model generation quality. However, non-diffusion approaches remain largely underexplored and it's unanswered whether AR models can be adapted as faster parallel decoders than dLLMs while maintaining generation quality. We present pcLLM, a progressive consistency distillation paradigm that transforms autoregressive (AR) models into efficient parallel decoders while preserving the causal inference property. pcLLM achieves $3.6\times$ wall-clock speedup on coding benchmarks with minimal loss in performance. Based on pcLLM's trajectory characteristics, we introduce multi-block decoding with rejection recycling, which enables up to $4.2\times$ higher token acceptance count per iteration and nearly $4\times$ speedup, effectively trading additional compute for lower inference latency.

## 1 INTRODUCTION

Modern Large Language Models (LLMs), such as GPT-5 (OpenAI, 2025), Gemini-2.5 (DeepMind, 2025), and DeepSeek-R1 (Ren et al., 2025), demonstrate impressive capabilities across a wide range of complex reasoning and agentic tasks. However, the strong performance come at the cost of high inference latency, particularly during the generation of long token sequences using chain-of-thought (Wei et al., 2022; Hou et al., 2025; Ren et al., 2025; Muennighoff et al., 2025) under autoregressive (AR) decoding. Since each token generation requires a full forward pass through the model, the sequential nature of decoding limits parallelism and underutilizes the massive parallel processing capabilities of modern GPUs. This results in significantly increased inference latency and high computational costs, degrading user experience in real-time and interactive applications.

Diffusion-based language models (dLLMs) offer an alternative to AR models by relaxing token-by-token causality and enabling multi-token generation per iteration with improved controllability (Li et al., 2024a; Nisonoff et al., 2024; Schiff et al., 2024). dLLMs reframe decoding as a more parallelizable computation that better utilizes the compute from modern accelerators.

Mercury (Inception Labs, 2025), Gemini Diffusion (Google DeepMind, 2025) and Seed Diffusion (Song et al., 2025b) demonstrate that diffusion-based LLMs (dLLMs) can achieve up to a $5\times$ increase in throughput while maintaining coding and text generation quality on par with autoregressive (AR) models. Community-driven efforts (Ye et al., 2025; Zhu et al., 2025; Nie et al., 2025a; JetAstra, 2025; Gong et al., 2025) are rapidly advancing in this direction; however, a performance gap remains. In particular, current open implementations often exhibit lower generation quality and face challenges in adapting widely used inference optimizations for AR models, such as KV caching, to the bi-directional attention setting of dLLMs. While recent work have made significant gains in further improving dLLMs' efficiency (Arriola et al., 2025; Wu et al., 2025b; Liu et al., 2025), it remains an open question whether AR models possess the same potential for parallel decoding, or the ability to train an efficient parallel decoder is a unique advantage of dLLMs.

One commonly used parallel decoding technique for AR models is Jacobi decoding (Song et al., 2021; Santilli et al., 2023), which is training-free and requires no architecture modification. While this method has inspired several extensions (Fu et al., 2024; Teng et al., 2024; Wu et al., 2025c), in practice these techniques deliver only modest speedups. Prior works including CLLM (Kou et al., 2024) and CEED-VLA (Song et al., 2025a) train LLMs and Vision-Language-Action (VLA) models

with consistency distillation (Song et al., 2023) to predict multiple correct tokens simultaneously in each iteration. Kou et al. (2024); Gat et al. (2025) observes that when inference with larger block sizes, the speedup achieved during inference plateau: as the block size increases, the number of tokens "fast-forwarded" per iteration remains essentially constant. A natural question is whether we can train models to better predict future tokens under Jacobi decoding, such that increasing the block size yields useful predictions. Modern AI accelerators offer high FLOPs, and if decoding more future tokens in each iteration could reduce the total number of iterations to generate the same number of tokens, total latency drops.

In this work, we introduce a progressive consistency distillation technique that address the limitation by progressively teaching to predict more tokens within each block and to perform better fast forwarding with increasing block size. We further introduce a noise-aware causal attention that teaches to model to predict correct tokens within each block conditioned on unconverged blocks, and we show it enables more useful future tokens to emerge in each block's trailing tails. We show applying rejection-recycling and multi-block decoding to leverage this model behavior from progressive consistency LLMs (pcLLM) for further efficiency improvement.

Experiments show pcLLM can serve as very efficient parallel decoders with up to $3.8\times$ improvement in generation speed across coding and math benchmarks. It also effectively generate higher quality draft n-grams from future tokens within each block, as observed in Section 4. Using rejection-recycling and multi-block decoding makes use of future n-grams and further boost speedup to $4.2\times$.

In summary, key contributions of this paper includes:

- We introduce progressive consistency distillation to train AR models as fast parallel decoders, pcLLM, with up to $4\times$ generation speedup.
- We empirically observe and qualitatively verify pcLLM have both higher fast-forwarded token count and a useful n-gram count in comparison with baseline models.
- We propose rejection-recycling and multi-block decoding to make use of higher quality draft n-grams from future tokens within each block, and apply them to pcLLM boost generation speed to $4.2\times$ across various benchmarks.

## 2 PRELIMINARY

This section reviews the basics of Jacobi decoding and consistency distillation training to accelerate Jacobi decoding of AR models.

### 2.1 JACOBI DECODING

Given a prompt $\boldsymbol{x}$ and a pre-trained LLM $p_\theta(\cdot|\boldsymbol{x})$ parametrized by $\theta$, the standard AR decoding under the greedy strategy produces a response sequentially as follows:

$$y_i = \arg\max_y p_\theta(y \mid \boldsymbol{y}_{<i}, \boldsymbol{x}), \quad \text{for } i = 1, \ldots, n, \tag{1}$$

where $\boldsymbol{y}_{<i} = \{y_1, \ldots, y_{i-1}\}$. This process requires $n$ forward passes of the LLM to generate $n$ tokens $\boldsymbol{y}_{\leq n}$. The inherently sequential nature of AR decoding limits practical efficiency when generating long sequences. Jacobi decoding (Song et al., 2021; Santilli et al., 2023) addresses this bottleneck by reformulating token generation as solving a system of nonlinear equations:

$$f(y_i, \boldsymbol{y}_{<i}, \boldsymbol{x}) = 0, \quad \text{for } i = 1, \ldots, n, \tag{2}$$

where $f(y_i, \boldsymbol{y}_{<i}, \boldsymbol{x}) := y_i - \arg\max_y p_\theta(y|\boldsymbol{y}_{<i}, \boldsymbol{x})$. This system can be solved in parallel using Jacobi fixed-point iteration (ort, 2000). Starting from a randomly initialized $n$-token sequence $\boldsymbol{y}^{(0)} = \{y_1^{(0)}, \ldots, y_n^{(0)}\}$, the update at each iteration $j$ is:

$$
\begin{cases}
y_1^{(j+1)} & = \arg\max_y p_\theta(y|\boldsymbol{x}) \\
y_2^{(j+1)} & = \arg\max_y p_\theta(y|\boldsymbol{y}_1^{(j)}, \boldsymbol{x}) \\
& \vdots \\
y_n^{(j+1)} & = \arg\max_y p_\theta(y|\boldsymbol{y}_{<n}^{(j)}, \boldsymbol{x}).
\end{cases}
\tag{3}
$$

Notably, for LLM, the above $n$ maximization problems can be solved in parallel by using a causal attention mask, i.e., only one forward pass of the LLM is required to obtain $\boldsymbol{y}^{(j+1)}$ based on $\boldsymbol{y}^{(j)}$. The iteration exits at some $k$ such that $\boldsymbol{y}^{(k)} = \boldsymbol{y}^{(k-1)}$ and we define $\boldsymbol{y}^* := \boldsymbol{y}^{(k)}$ as the fixed point. Let $\mathcal{J} := \{\boldsymbol{y}^{(0)}, \ldots, \boldsymbol{y}^{(k)}\}$ denote the Jacobi trajectory. It can be proven that $\boldsymbol{y}^*$ is identical to AR decoding under greedy strategy (Song et al., 2021).

To generate a long response $\boldsymbol{l}$ of length $L \gg n$, Jacobi decoding is applied sequentially over blocks of size $n$ until the `<eos>` token appears in a fixed point. Let $\boldsymbol{y}_{B_i}^*$ denote the fixed point obtained for the $i$-th block. The full output $\boldsymbol{l}$ is then constructed by concatenating fixed points from consecutive blocks:

$$\boldsymbol{l} = [\boldsymbol{y}_{B_1}^*, \ldots, \boldsymbol{y}_{B_N}^*], \tag{4}$$

where $N = \lceil \frac{L}{n} \rceil$ denotes the number of blocks generated before termination.

## 2.2 CONSISTENCY DISTILLATION

Despite the promise, Jacobi decoding achieves little speedup over standard AR decoding (Santilli et al., 2023; Fu et al., 2024), as it rarely predicts more than one correct[1] token within one fixed-point iteration. To address this, recent works such as CLLMs (Kou et al., 2024) propose consistency distillation, a training approach designed to accelerate convergence to the fixed point from arbitrary states on a Jacobi trajectory. The key idea is to introduce a consistency loss that encourages an LLM $p_\theta(\cdot|\boldsymbol{x})$ to predict multiple tokens simultaneously:

$$\mathcal{L}_c = \mathbb{E}_{i \sim \mathcal{U}\{1,\ldots,N\}, \boldsymbol{y}_{B_i} \sim \mathcal{J}_i} \left[ D_{\text{KL}} \left( p_{\theta^-}(\boldsymbol{y}_{B_i}^*|\boldsymbol{x}, \boldsymbol{y}_{B_1}^*, \ldots, \boldsymbol{y}_{B_{i-1}}^*) || p_\theta(\boldsymbol{y}_{B_i}|\boldsymbol{x}, \boldsymbol{y}_{B_1}^*, \ldots, \boldsymbol{y}_{B_{i-1}}^*) \right) \right], \tag{5}$$

where $\theta^- = \text{stopgrad}(\theta)$ and $D_{\text{KL}}$ denotes the KL divergence aggregated across the $n$ tokens in a block. Here, $i \sim \mathcal{U}\{1, \ldots, N\}$ denotes sampling a block index uniformly at random, and $\boldsymbol{y}_{B_i} \sim \mathcal{J}_i$ denotes randomly sampling from the Jacobi trajectory of the $i$-th block.

CLLMs build upon this idea by first collecting Jacobi trajectories, obtained by running Jacobi decoding with $p_\theta$ on a set of prompts. The model is then trained with a joint objective that combines the consistency loss in Eq. 5 with the standard AR loss, achieving up to a $2\times$ speedup over AR decoding while maintaining quality. Similar training objectives have also been adopted for inference acceleration in other domains, such as action prediction in VLA models (Song et al., 2025a).

## 3 METHODOLOGY

In this section, we first discuss the training challenges of consistency distillation with larger block sizes $n$, and then present progressive consistency distillation, a refined paradigm designed to mitigate this bottleneck, and denote LLMs trained under this paradigm as pcLLM. Furthermore, by observing pcLLM's trajectories under vanilla Jacobi decoding, we introduce rejection-recycling and multi-block decoding strategies to improve its efficiency.

### 3.1 PROGRESSIVE CONSISTENCY DISTILLATION

**Progressive Noise Schedule.** In Jacobi decoding, we maintain strict causality within each block, where each token is updated in accordance with Eq. 3. Consider the $i$-th block $\boldsymbol{y}_{B_i}^{(j)}$ of size $n$ is been decoded at some iteration step $j$. Assume the first $c - 1$ tokens have been accepted, and we denote $y_f$ as the future token as shown in Eq. 6.

$$y_f = \arg\max_y p\left(y \mid \boldsymbol{x}_c, \boldsymbol{y}'_{c:f-1}\right), \quad \text{for } f = c+1, \ldots, n, \tag{6}$$

where $\boldsymbol{x}_c = [\boldsymbol{x}, \boldsymbol{y}_{<c}]$ is the clean context, $\boldsymbol{y}'_{c:f-1}$ is the noisy[2] context. While the training objective in Eq. 5 is designed to optimize correct token prediction in this setting, it's observed from Kou et al.

---

[1] By correctness, we mean alignment with the AR decoding result under a greedy sampling strategy.

[2] By noisy, we refer to tokens in the non-converged point along the Jacobi trajectory that that differ from those in the fixed point at the same positions.

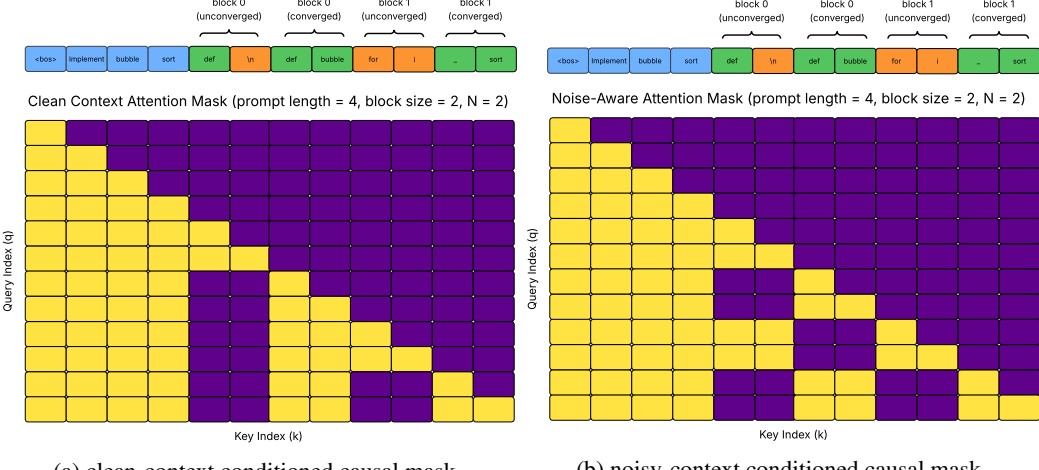

(a) clean-context conditioned causal mask.  (b) noisy-context conditioned causal mask.

Figure 1: Sequence packing with two attention mask implementations, both allow logits from clean blocks and noisy blocks to be generated with single forward pass to calculate the progressive consistency loss and AR loss in Eq. 9.

(2024) that predicting $y_f$ is hard when it's conditioned on a long noisy context $\boldsymbol{y}'_{c:f-1}$ under large block sizes (e.g., $n = 256$).

To address this challenge, we instead split a large block into smaller blocks (e.g., $n = 16$) with noise ratios determined by a predefined schedule $\{t_1, \ldots, t_N\}$. Each $t_i$ denotes the fraction of noisy tokens in a block. The noise schedule follows a cyclic strategy with window size $w$, where the noise ratio linearly increases from 0 to 1 within each window, i.e.,

$$W = \left\{0, \frac{1}{w}, \ldots, \frac{w-1}{w}\right\}, \quad t_i = W[j], \quad j = i \bmod w. \tag{7}$$

This progressive schedule ensures that each block retains a partially clean context, thereby shortening noisy tokens dependencies. In particular, it reduces the longest span of consecutive noisy inputs for any prediction from $O(nN)$ assuming $t_i = 1$ for all blocks using random schedule to $O(\lceil tn \rceil)$ using progressive schedule, which facilitates learning. Empirically, we find this progressive schedule to be more effective than a purely random noise schedule (Table 4).

**Progressive Distillation Loss.** Let $\boldsymbol{y}^{t_i}_{b_i}$ denote the point along the $i$-th block Jacobi trajectory with a number of noisy tokens closest to $\lceil t_i n \rceil$. The training objective is to predict tokens correctly within each block, aggregating losses across blocks to reduce gradient variance and stabilize optimization. Accordingly, we introduce a new loss term, *progressive consistency loss*, which optimizes $p_\theta$ under the progressive noise schedule in Eq. 7:

$$\mathcal{L}_{\text{pc}} = \frac{1}{N} \sum_{i=1}^{N} D_{\text{KL}}\left(p_{\theta^-}(\boldsymbol{y}^*_{B_i} \mid \boldsymbol{x}, \boldsymbol{y}^{t_1}_{B_1}, \ldots, \boldsymbol{y}^{t_{i-1}}_{B_{i-1}}) \,\middle\|\, p_\theta(\boldsymbol{y}^{t_i}_{B_i} \mid \boldsymbol{x}, \boldsymbol{y}^{t_1}_{B_1}, \ldots, \boldsymbol{y}^{t_{i-1}}_{B_{i-1}})\right). \tag{8}$$

**AR Loss.** Kou et al. (2024) notes that using only the consistency loss (Eq. 5) must be supplemented with an AR loss to maintain generation quality. Our preliminary experiments show that using only the consistency objective (Eq. 8) produces the same effect. This motivates our inclusion of a conventional AR loss term in the final training objective to safeguard output quality:

$$\mathcal{L}(\theta) = \mathcal{L}_{\text{pc}} + w\mathcal{L}_{\text{AR}} \tag{9}$$

where $w$ is a tunable weight that balances the two learning objectives.

**Noise-aware Causal Attention.** In CLLM, loss from each training step is computed based on KL divergence from one block instance in Eq. 5. This learning objective is to train correct token prediction in the setting where there is only a big block (Eq. 6). Moreover, in both Eq. 5 and Eq. 8, the loss term computation involves two forward passes using a conventional causal mask since each involves a distinction sequence. As a result, it requires $O(2N)$ forward passes to compute all loss terms in Eq. 8 and $O(N)$ backward passes to compute gradients, resulting in low training efficiency,

| | accepted tokens | | noisy tokens | | fixed point segments | | | | | | | | | | | | | | | | | |
|---|---|---|---|---|---|---|---|---|---|---|---|---|---|---|---|---|---|---|---|---|---|---|
| 1: | 73594 | ...... | 12128 | 311 | 1817 | 1008 | 1817 | 262 | 262 | 2661 | 2661 | 2661 | 624 | 262 | 702 | 12704 | 22801 | 2561 | 13 | 13 | 13 | 55722 | ...... |
| 2: | 73594 | ...... | 12128 | 311 | 1817 | 1008 | 1091 | 198 | 262 | 262 | 12171 | 2661 | 624 | 262 | 12109 | 262 | 702 | 2561 | 16 | 13 | 15 | 11 | ...... |
| 3: | 73594 | ...... | 12128 | 311 | 1817 | 1008 | 1091 | 198 | 262 | 2661 | 12171 | 624 | 262 | 262 | 12109 | 702 | 12704 | 22081 | 2561 | 16 | 13 | 15 | ...... |
| 4: | 73594 | ...... | 12128 | 311 | 1817 | 1008 | 1091 | 198 | 262 | 2661 | 12171 | 624 | 262 | 12109 | 702 | 12704 | 22081 | 22081 | 16 | 16 | 13 | 15 | ...... |
| 5: | 73594 | ...... | 12128 | 311 | 1817 | 1008 | 1091 | 198 | 262 | 2661 | 12171 | 624 | 262 | 12109 | 702 | 12704 | 22801 | 2561 | 16 | 13 | 15 | 11 | ...... |
| 6: | 73594 | ...... | 12128 | 311 | 1817 | 1008 | 1091 | 198 | 262 | 2661 | 12171 | 624 | 262 | 12109 | 702 | 12704 | 22801 | 2561 | 16 | 13 | 15 | 11 | ...... |

*Rejection-recycling helps me solve faster!*

Figure 2: Visualization of pcLLM's trajectory under vanilla Jacobi decoding. The figure shows a partial segment of the trajectory. Blue tokens denote accepted tokens that match the fixed point at their positions. Black tokens denote unconverged noisy tokens, and we highlight them in red if more than three consecutive tokens match the fixed point regardless of position.

especially in settings like CoT generation for reasoning models. We reduce the number of forward and backward passes from $O(N)$ to $O(1)$ by introducing a sequence packing technique and a block-wise sparse attention mask. We illustrate the sequence packing that interleaves $\boldsymbol{y}_{b_i}^{t_i}$ and $\boldsymbol{y}_{b_i}^*$ for the entire complete sequence in Figure 1b for $\mathcal{L}_{pc}$ computation, in contrast with conditioning each unconverged $y_{b_s}$ only on clean tokens for consistency distillation with $\mathcal{L}_c$ in Figure 1a.

**Progressive Distillation for Larger Block Sizes.** In training pcLLM on Jacobi trajectories prepared from the original AR model, we find model speedup scales with the number of training steps and saturate at around 400k steps. We find that collecting additional rounds of Jacobi trajectories from intermediate checkpoints empowered with multi-token prediction capability and train the models on new trajectories with *progressively larger block sizes* can break the ceiling and further improve model speedup by up to $20\%$, yet with a slight degradation of model performance.

### 3.2 INFERENCE OPTIMIZATION

**Jacobi Decoding Behavior of pcLLM.** pcLLM is trained to have a stronger capability of generating correct future tokens conditioning on noisy tokens. Qualitative analysis in Figure 2 illustrates that it indeed brings the quality improvement: fixed-point segments emerge within the noisy tokens of the unconverged point. Furthermore, these segments progressively extend (e.g., the number of red tokens increases from point 1 to point 2 in Figure 2), even under noisy context, consistent with our training patterns. In this section, we focus on how to translating this qualitative observation of draft quality improvement into qualitative speedup.

**Rejection Recycling.** Prior work has shown that n-grams produced during Jacobi iterations can be verified in parallel and reused in subsequent iterations (Fu et al., 2024). As illustrated in Figure 2, such n-gram sizes could be large in pcLLM, and if correctly verified many tokens can be fast-forwarded in one iteration. In particular, we initialize a fixed-size n-gram pool by collecting noisy token sequences from unconverged points during Jacobi decoding. If the pool contains an n-gram matching the last accepted token of the current point, we concatenate its subsequent tokens to form new candidates (line 11 in Algorithm 1). At each iteration, we select the candidate with the most accepted tokens. For instance, this strategy enables skipping from point 3 to point 5 in Figure 2, as the fixed-point segments in point 3 yield higher-quality candidates.

**Multi-block Decoding.** In addition to high-quality n-grams in the draft, we also observe the increasing number of stationary tokens, which are correctly predicted with preceding noisy tokens and remain unaltered through subsequent iterations. Together they yield higher quality drafts. To make use of the property, we introduce *multi-block decoding*, a new decoding paradigm that maintains and refines up to $K$ blocks simultaneously. It marks the block closest to the effective KV cache boundary as the *real-active* block and all the other $K-1$ blocks as *pseudo-active* blocks. Only tokens within the real-active block are accepted and committed to KV cache. Tokens in pseudo-active blocks are only pseudo-accepted, conditioning on prior blocks; once converged, pseudo-

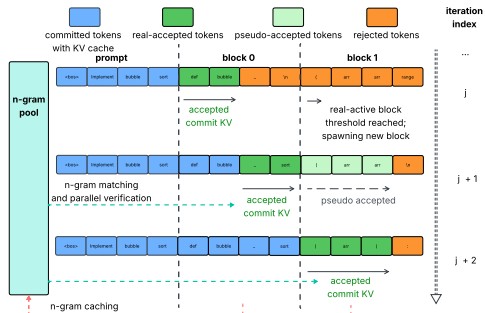

Figure 3: An example of multiblock decoding with rejection recycling at prompt length = 4, block size = 4, $r = 0.5$, $K = 2$.

---
**Algorithm 1** MULTIBLOCK DECODING + REJECTION RECYCLING

---
1: **Init:** Create a set of blocks $\{b\}$ with one *real–active* block $RA$: draft tokens $q_{RA}$ randomly initialized, accepted tokens $a_{RA} = \varnothing$ ; For all other blocks $b$, set $q_b = \varnothing$, $a_b = \varnothing$, and mark as *pseudo-active*.
2: Initialize candidate pool $\mathcal{N} = \emptyset$, spawn ratio $r$, threshold $s = \lceil rn \rceil$, block size $n$.
3: **while** iters < max **do**
4:     **Assemble input $y$:** Concatenate $q_{RA}$, then for each pseudo-active $b$, append $a_b$ (no logits) and $q_b$ (collect logits). Resize cache to batch $y$.
5:     **Forward:** Run model $p_\theta(y)$ to produce logits.
6:     **for** each block $b$ with span $(start, L)$ **do**
7:         **Verification (with rejection-recycling):** Greedy prediction $g = \arg\max$ logits; accept longest matching prefix of $q_b$ using $g$ (or $g \cup \mathcal{N}$ if $b = RA$); update $a_b$.
8:         **if** $b = RA$ and EOS encountered in accepted region **then**
9:             **return** committed output.
10:         **end if**
11:         **Tail update:** If partial accept, set $q_b \leftarrow [\text{next} \| g_{\text{tail}}]$ (and if $b = RA$: push rejected tail to update $\mathcal{N}$ and $q_{RA}$); else $q_b \leftarrow \varnothing$.
12:     **end for**
13:     **Cache trim:** Delete false KV to committed length: prompt + verified $a_b$ (all accepted blocks) + $a_{RA}$.
14:     **Spawn:** If some block $b$ reaches $|a_b| \geq s$ and active $\{b\} < K$, clone and pad $q_{RA}$ to length $n$ and add as new pseudo-active block.
15:     **Promote:** If $|a_{RA}| \geq n$, choose a pseudo-active $b$ with $|a_b| > 0$, rebuild its draft to length $n$, mark as verified, set $RA \leftarrow b$.
16:     **Stop:** If all $|a_b| \geq n$ or EOS emitted by $RA$, break.
17: **end while**
18: **Finalize:** Concatenate output = verified $a_b$ for all non-RA blocks, then $a_{RA}$; trim KV cache $\mathcal{C}$;
19: **Return:** (output, $\mathcal{C}$, iters)

---

active blocks will wait until they are promoted as the
real active block, where all tokens will be verified again, but now with a higher-quality draft. A detailed description is provided in Algorithm 1 (with rejection recycling) and with an example in Figure 3. Note that both rejection recycling and multi-block decoding are lossless as they employ greedy rejection sampling for token acceptance in the real-active block (Leviathan et al., 2022).

## 4 EXPERIMENTS

### 4.1 EVALUATION SETTINGS

**Models and Datasets.** We evaluate pcLLM across coding benchmark. For coding benchmarks, we train Qwen2.5-Coder-Insutrct (Hui et al., 2024) on OpenCodeInstruct (Ahmad et al., 2025) and test on the HumanEval (Chen et al., 2021), MBPP (Austin et al., 2021b). On OpenCodeInstruct, we curate question instances that come with generations that pass all unit tests, from where we use 450k prompts for trajectory generation and training. For mathematical tasks, we train Qwen2.5-Math-7B-Instruct (Yang et al., 2024) on the math split of Openthought2 (Guha et al., 2025) and test on GSM8K (Cobbe et al., 2021), and MATH (Hendrycks et al., 2021). On Openthought2, only mathematical prompts are considered, from where we apply the same training settings for trajectory generation and training.

**Training Settings**. All training and inference are conducted on instances equipped with 8x NVIDIA A100-80GB GPUs, and 8x NVIDIA H200 GPUs. All models are trained with learning rate lr=$10^{-6}$, max new sequence length at 2048. For pcLLM, we adopt linear progressive noise schedule, initial block size at 16, window size at 16, and a second round of trianing with block size at 32, window size at 8. Ablation studies on parameter choices are presented in Section 4.3.

**Baselines.** Our main objective in this section is to compare performance and efficiency between diffusion-based parallel decoders and AR-based parallel decoder, pcLLM. The dLLM baselines, also

Table 1: Performance and efficiency on coding benchmarks, HumanEval and MBPP, grouped by decoding family. For AR-based models, all methods adopt Qwen2.5-Coder-7B-Instruct. For pcLLM, MR stands for employing the multi-block and rejection-recycling decoding algorithm introduced in Algorithm 1. DC stands for using bi-directional dual cache from fast-dLLM. For both Fast-dLLM and D2F, we choose the Dream-7B as it's significantly faster with similar or better performance than LLaDA-7B. For CLLM*, we follow mostly the same recipe in CLLM but with new sequence packing technique (without progressive training on larger block sizes). The speedup ratio is relative to the AR baseline.

| Benchmark | Family | Method | TPS ↑ | Speedup ↑ | Accuracy ↑ | Param Size (B) |
|---|---|---|---|---|---|---|
| **HumanEval** | *AR-based* | AR | 41.3 | 1.00× | 87.8 | 7 |
| | | Jacobi | 39.9 | 0.97× | 87.8 | 7 |
| | | CLLM* | 103.3 | 2.50× | 87.8 | 7 |
| | | pcLLM | 147.6 | 3.57× | 84.8 | 7 |
| | | pcLLM (MR) | **149.3** | **3.62×** | 84.8 | 7 |
| | *Diffusion-based* | LLaDA-Instruct | 2.8 | 0.07× | 36.0 | 7 |
| | | Dream-Base | 20.2 | 0.49× | 54.3 | 7 |
| | | Fast-dLLM (DC) | 60.0 | 1.45× | 53.0 | 7 |
| | | D2F | 73.2 | 1.77× | 54.3 | 7 |
| **MBPP** | *AR-based* | AR | 43.1 | 1.00× | 74.3 | 7 |
| | | Jacobi | 42.4 | 0.98× | 74.3 | 7 |
| | | CLLM* | 80.1 | 1.94× | 71.4 | 7 |
| | | pcLLM | 90.4 | 2.10× | 73.4 | 7 |
| | | pcLLM (MR) | **106.3** | **2.47×** | 73.4 | 7 |
| | *Diffusion-based* | LLaDA-Instruct | 0.9 | 0.02× | 39.0 | 7 |
| | | Dream-Base | 10.4 | 0.24× | 56.2 | 7 |
| | | Fast-dLLM (DC) | 73.2 | 1.70× | 51.0 | 7 |
| | | D2F | 105.0 | 2.44× | 55.2 | 7 |

have the capability of generating a single block of tokens or multiple consecutive blocks of tokens together. Specifically, we compare pcLLM with state-of-the-art (SOTA) dLLMs including LLaDA-7B (Nie et al., 2025b), Dream-7B (Ye et al., 2025), fast-dLLM (Wu et al., 2025b) and D2F (Wang et al., 2025). We also compare pcLLM with AR-based parallel decoder including vanilla Jacobi decoding (Santilli et al., 2023) and CLLM (Kou et al., 2024). In addition, to situate pcLLM among broader AR-acceleration techniques, we present in the appendix a complementary comparison with speculative decoding methods, including EAGLE-3 (Li et al., 2025) and HASS (Zhang et al., 2025), and with more recent dLLM baselines such as Fast-dLLM v2 (Wu et al., 2025a), SDAR (Cheng et al., 2025), and the consistency-distilled dLLM dParallel (Chen et al., 2025).

## 4.2 RESULTS

**Performance.** The performance metrics are the greedy generations' strict accuracy (pass@1) on HumanEval and MBPP. Table 1 compares pcLLM with both dLLMs and Jacobi decoding baselines. On A100 GPUs, our results show that on both benchmarks, pcLLM consistently achieves competitive accuracy with a much better speedup at the same parameter scale. In particular, for structured generations like Python coding, pcLLM achieves 3.6× speedup in com-

Table 2: Speedup on HumanEval tested on H200 using same settings and speedup ratio over A100.

| Method | TPS ↑ | Speedup↑ | r (vs. A100) |
|---|---|---|---|
| AR | 65.4 | 1.00× | 1.00 |
| Jacobi | 63.0 | 0.96× | 1.00 |
| CLLM* | 151.3 | 2.31× | 0.92 |
| pcLLM | 235.3 | 3.60× | 1.01 |
| pcLLM (MR) | **258.3** | **3.95×** | 1.09 |

parison with the AR baseline, $7.4 \sim 53.3\times$ speedup comparing to dLLM baselines, and $2.0\times$ comparing to optimized dLLM baselines including Fst-dLLM and D2F with techniques like adding block-wise KV cache, bidirectional KV cache and pipelined parallel decoding. For speedup evaluation, we run all evaluations with block size at 128 except for pcLLM (MR) since MR takes extra FLOPs for multiblock decoding and parallel verification. We also present speedup comparison across different AR-based techniques with pcLLM on H200 in Table 2 as it comes with a better fast-forward count to TPS conversion rate with mroe compute on H200. On H200, with the block size at 64 and verification size at 4 (rationale provided in Section 4.3), we apply multi-block de-

Table 3: Performance and efficiency on math benchmarks, GSM8K and MATH, grouped by decoding family. For AR-based models, all methods adopt Qwen2.5-Math-7B-Instruct.

| Benchmark | Family | Method | TPS ↑ | Speedup ↑ | Solve Rate ↑ | Param Size (B) |
|---|---|---|---|---|---|---|
| **GSM8K** | *AR-based* | AR | 41.8 | 1.00× | 92.4 | 7 |
| | | Jacobi | 43.8 | 1.05× | 92.4 | 7 |
| | | CLLM* | 86.8 | 2.08× | 92.2 | 7 |
| | | pcLLM | 146.1 | 3.50× | 91.4 | 7 |
| | | pcLLM (MR) | **154.9** | **3.71×** | 91.4 | 7 |
| | *Diffusion-based* | LLaDA-Instruct | 7.2 | 0.17× | 77.4 | 7 |
| | | Dream-Base | 9.5 | 0.23× | 75.0 | 7 |
| | | Fast-dLLM (DC) | 49.8 | 1.19× | 75.0 | 7 |
| | | D2F | 91.2 | 2.18× | 77.6 | 7 |
| **MATH** | *AR-based* | AR | 41.3 | 1.00× | 77.0 | 7 |
| | | Jacobi | 42.2 | 1.02× | 77.0 | 7 |
| | | CLLM* | 84.4 | 2.04× | 77.2 | 7 |
| | | pcLLM | 150.7 | 3.65× | 77.4 | 7 |
| | | pcLLM (MR) | **152.0** | **3.68×** | 77.4 | 7 |
| | *Diffusion-based* | LLaDA-Instruct | 21.1 | 0.51× | 23.7 | 7 |
| | | Dream-Base | 9.9 | 0.24× | 35.8 | 7 |
| | | Fast-dLLM (DC) | 67.0 | 1.62× | 37.1 | 7 |
| | | D2F | 98.8 | 2.39× | 35.4 | 7 |

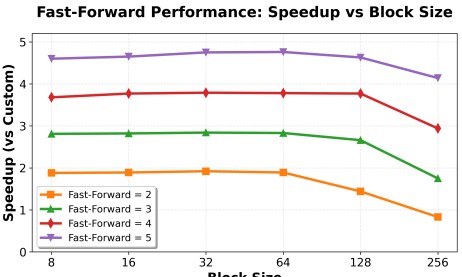

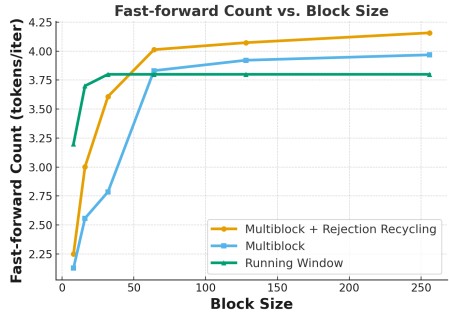

(a) Speedup vs. (log-scaled) block size at fixed fast-forwarding count per iteration on NVIDIA H200 GPU, using Jacobi decoding at prompt length = 128, generation length = 256.

(b) fast-forward count vs. block size on HumanEval using three decoding strategies on NVIDIA H200 GPU. Notice larger block size provides more fast-forward token count for multi-block decoding with rejection recycling.

Figure 4: Effect of block size choices on fast-forward counts and wall-clock speedup under different settings. We choose the maximum block size on hardware without sacrificing wall-clock speedup.

coding using pcLLM and the results are presented in Figure 3. The running window method is an optimized variant of Jacobi decoding designed for settings where many tokens are accepted per iteration. It maintains a fixed-size active block by replenishing draft tokens to the original block size as accepted tokens are committed to the KV cache. The results demonstrate that multi-block decoding with rejection recycling consistently achieves the highest number of fast-forwarded tokens per iteration, particularly in the larger block-size regime as shown in Figure 4b.

## 4.3 ABLATION STUDY

**Training Noise schedules.** We evaluate three types of noise schedules: random, linear progressive, and reverse progressive. In the random schedule, the noise step $t_i$ for each block is sampled uniformly as $t_i \sim \mathcal{U}(1, \ldots, N)$ during sequence packing in pcLLM training. The linear progressive schedule follows Eq. 7, while the reverse progressive schedule applies a linearly decreasing noise ratio from 1 to 0 within each window. Results in Table 4 show that the linear progressive schedule significantly outperforms the other two when the window size is 8. Intuitively, with $N = 16$, this schedule corresponds to adding noise more aggressively across blocks within each window, roughly two additional noisy tokens per future block, until the final block where all tokens are noisy.

Table 4: Inference results for block size = 256 with $N = 16$, $t_{\min} = 0.0$ and $t_{\max} = 1.0$. Acc. = pass@1 accuracy (%) on HumanEval. The checkpoints are trained with Qwen2.5-Coder-7B-Instruct on 10k randomly sampled instances from our OpenCodeInstruct trajectory dataset. Notice that for ablation purpose, the checkpoints are not trained with full datascale as in Table 1. Reverse progressive is significantly worse than other schedule and we only conduct ablation for one choice of window size.

| Window Size | Random | | Linear Progressive | | Reverse Progressive | |
|---|---|---|---|---|---|---|
| | Acc. | iter/token | Acc. | iter/token | Acc. | iter/token |
| 8 | 82.9 | 0.53 | **84.7** | **0.48** | – | – |
| 16 | 83.5 | 0.51 | 81.7 | 0.46 | 82.9 | 0.62 |

**Training Mask types.** We train pcLLM on the objective in Eq. 8 with noise-conditioned mask implementation (Figure 1b). An alternative implementation of the mask is to condition all blocks within a window on clean context. In other words, for every query, it sees blocks from all proceeding windows as of Figure 1a]), and all blocks within its own window as of Figure 1b. Intuitively, it makes token predictions in later windows and blocks easier to learn because now

Table 5: Effects of applying noise-conditioned mask (NC) or noise-conditioned mask with intra-window clean context (NC-IC) for pcLLM training, and evaluated on HumanEval with A100.

| Method | Speedup↑ | Acc. |
|---|---|---|
| NC | **3.6×** | **82.3** |
| NC-IC | 1.9× | 82.3 |

they are conditioned on cleaner context. We summarize results in Table 5, where it shows noise-conditioned mask is more effective in empowering pcLLM with speedup while maintaining generation quality.

**Inference FLOPs Utilization Analysis.** pcLLM (MR) involves both multi-block decoding and rejection-recycling, each technique consumes extra FLOPs for parallel drafting and parallel verification respectively. To maximize hardware utilization, we experiment with how end-to-end decoding latency changes as the total number of decoded tokens changes. We use Jacobi decoding to run the experiments and the results are shown in Figure 4a. On H200 GPUs, Jacobi decoding with block sizes up to 64 shows no latency penalty and only minor degradation at 128, particularly in the high fast-forwarding regime. The result is consistent across accepted token counts fixed at $2, 3, 4, 5$, indicating that up to 126 tokens can be decoded in parallel with shared KV without significant latency overhead.

**Inference Configuration Search.** Beyond block size, the main tunable parameters for pcLLM (MR) inference are verification size (entries verified in parallel with shared KV for rejection recycling), number of blocks, and initialization threshold. Performance gains from additional blocks saturate at block size = 2 as later drafts degrade quickly. The initialization threshold, defined as the fraction of the first block completed before launching the next, can be optimized via grid search and shows consistently optimal performance at $r = 0.85$ for block size 64 across verification sizes 2 to 8. For maximum FLOPs utilization, we use block size = 64, verification size = 4, where wall-clock speedup remains stable until parallel decoding exceeds 256 tokens.

## 5 RELATED WORK

**Discrete Text Diffusion.** Early milestones in discrete diffusion for language modeling include D3PM for discrete state spaces (Austin et al., 2021a), SEDD, which uses a score-entropy loss for competitive discrete text generation (Lou et al., 2024), masked diffusion language models MDLM and MD4, which simplify and generalize masked discrete diffusion for text (Sahoo et al., 2024; Shi et al., 2024), and RADD, a reparameterized discrete diffusion model that unifies absorbing diffusion with any-order AR models while enabling faster sampling (Ou et al., 2025). . Building on these foundations, diffusion large language models (dLLMs) have emerged as a new paradigm that challenges traditional autoregressive (AR) modeling by replacing left-to-right causality with iterative denoising, enabling parallel multi-token generation (Li et al., 2024a; Nisonoff et al., 2024; Schiff et al., 2024). Closed-source dLLMs (e.g., Gemini Diffusion (Google DeepMind, 2025; Inception

Labs, 2025; Song et al., 2025b)) show huge throughput improvement while maintaining competitive code and text quality, underscoring better accelerator utilization. On the open-source side, community dLLMs with released code and weights delivered strong throughput and controllability via parallel iterative denoising, yet remaining less efficient than autoregressive decoding (Ye et al., 2025; Zhu et al., 2025; Nie et al., 2025a; JetAstra, 2025; Gong et al., 2025). Recent efforts (Arriola et al., 2025; Wu et al., 2025b; Liu et al., 2025) further push the efficiency and scalability of dLLMs.

**Speculative Decoding**. Speculative decoding speeds up AR generation by letting a lightweight drafter propose several future tokens and having the target model verify them in one pass (Leviathan et al., 2022; Chen et al., 2023). It preserves the target model's distribution while reducing latency. Subsequent work improves proposal quality and verification efficiency: online speculative decoding (OSD) (Liu et al., 2024) adapts draft models to user query distributions via continual distillation, substantially improving token acceptance and reducing inference latency. Medusa (Cai et al., 2024) adds multi-head drafters to the base LM to produce verify-able token blocks; EAGLE, EAGLE-2 (Li et al., 2024b;c) reuse target features for feature-level drafting, and EAGLE-3 (Li et al., 2025) scales this idea with multi-layer fusion. Lookahead Decoding (Fu et al., 2024), PLD (Saxena, 2023; Somasundaram et al., 2024), and REST (He et al., 2023) dispense with a separate drafter, instead synthesizing speculative candidates directly from context or future tokens. The self-speculative decoding paradigm shares a close connection with the Jacobi decoding adopted in this work.

**Jacobi Decoding.** Jacobi decoding reframes AR generation as a parallel fixed-point update over all positions, with convergence linked to greedy AR, and has been instantiated using Jacobi (Gauss-Seidel) iterations (Song et al., 2021; Santilli et al., 2023). Building on this, follow-ups either refine the decoding procedure or train models as parallel decoders to exploit parallelism at inference time: CLLMs (Kou et al., 2024) fine-tune LLMs with consistency distillation to predict multiple correct tokens per iteration and speed convergence; CEED-VLA (Song et al., 2025a) brings the similar idea to robotics. Other strands adapt Jacobi to new regimes, including FastCoT (Zhang et al., 2023) for reasoning with parallel CoT updates, Speculative Jacobi Decoding (Teng et al., 2024) for sampling in AR Test-to-Image, and MSN, TR-Jacobi (Wang et al., 2024) that injects denoising training and a retrieval-augmented Jacobi strategy.

# 6 CONCLUSION

In this work, we propose a progressive distillation technique for training AR models as faster and more accurate parallel decoders compared to dLLMs. Unlike CLLM (Kou et al., 2024), which directly trains models to predict large blocks of tokens in parallel, our approach introduces a progressively more difficult learning objective. This is achieved through a progressive noise schedule, combined with a sequence packing strategy and a noise-aware causal mask, enabling parallel token prediction conditioned on noise. The model is further improved through iterative training, where trajectories are regenerated with progressively larger block sizes. The resulting model, pcLLM, achieves a $3.6\times$ speedup while largely preserving accuracy. Analysis of its generated trajectories shows that pcLLM produces high-quality draft tokens toward the tail of sequences. In addition, we introduce rejection recycling and multi-block decoding, which together brings tokens accepted per iteration to $4.2\times$ as high with nearly $4\times$ speedup on HumanEval using an H200 GPU.

ETHICS STATEMENT

All authors have read and adhere to the ICLR Code of Ethics. This work does not involve human subjects, sensitive personal data, or experiments with the potential to cause harm. No confidential or proprietary data were used. The methods and experiments were conducted in accordance with principles of research integrity, fairness, and transparency. Potential societal impacts, including limitations and biases of large language models, are explicitly discussed in the paper. All conclusions are the sole responsibility of the authors.

REPRODUCIBILITY STATEMENT

We have made significant efforts to ensure the reproducibility of our results. Detailed descriptions of the models, datasets been used, as well as hyperparameter choices are included in the main text. All datasets used are publicly available, and the preprocessing steps are fully documented. Ablation studies are provided to validate robustness of results. These resources collectively allow independent researchers to verify and reproduce our work.

## 7 USE OF LLM

During the preparation of this manuscript, large language model was used to refine grammar and improve clarity. The authors carefully reviewed and revised all outputs to ensure the text reflects their original ideas and take full responsibility for the final content, including all statements and conclusions.

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

# A    FURTHER BASELINE COMPARISONS

The main text focuses on comparisons between pcLLM and diffusion-based parallel decoders, as well as AR-based parallel decoders, under a controlled setup where AR variants share the same backbone (Qwen2.5-Coder-7B-Instruct). This appendix extends the comparison to (i) distilled discrete diffusion models and (ii) state-of-the-art speculative decoding baselines.

Table 6: Additional comparison on HumanEval across AR, speculative decoding, and dLLM-based methods. For the AR baseline and all Jacobi-decoding based methods, Qwen2.5-Coder-7B-Instruct is used as the backbone. Speedup is measured in TPS relative to the AR baseline on a single B200 GPU.

| Family | Method | Acc. ↑ | TPF ↑ | TPS ↑ | Speedup vs. AR ↑ |
|---|---|---|---|---|---|
| AR | AR (greedy) | 87.8 | 1.00 | 83.00 | 1.00× |
| dLLM | Fast-dLLM v2 | 63.4 | 1.00 | 83.29 | 1.00× |
| dLLM | SDAR | 78.7 | 2.36 | 31.46 | 0.38× |
| dLLM (distilled) | dParallel | 54.3 | 2.90 | 175.15 | 2.11× |
| AR + Spec-Dec | EAGLE-3[*] | 68.9[*] | 6.38 | 246.10 | 2.97× |
| AR + Spec-Dec | HASS[*] | 61.6[*] | 5.53 | 280.29 | 3.37× |
| AR + Jacobi | Jacobi | 87.8 | 1.05 | 84.70 | 1.02× |
| AR + Jacobi | CLLM | 87.8 | 2.80 | 207.40 | 2.50× |
| AR + Jacobi | pcLLM | 84.8 | 3.96 | 299.50 | 3.61× |
| AR + Jacobi | pcLLM (MR) | 84.8 | 4.21 | **319.57** | **3.85×** |

[*]EAGLE-3 and HASS use different base models (DeepSeek-R1-Distill-Llama-8B and LLaMA3-Instruct, respectively) and are therefore not strictly comparable to the Qwen2.5-7B backbone, but they reflect the strongest checkpoints released by the authors.

**Distilled dLLM baselines.**    A distilled dLLM baseline is useful for mapping pcLLM against contemporary training techniques for discrete diffusion models. dParallel (Chen et al., 2025) performs trajectory-level consistency distillation on a discrete diffusion model to accelerate token sampling while aiming to preserve quality. We adopt the technique as the latest distilled dLLM baseline.

As shown in Table 6, on HumanEval, pcLLM (MR) attains a noticeably stronger speed–quality profile than dParallel: pcLLM (MR) achieves 29% higher accuracy and achieves more than 80% higher TPF and TPS. On GSM8K, pcLLM improves accuracy by 8 absolute points with about 20% higher TPF and TPS (GSM8K numbers are omitted from the table below for brevity). These gaps indicate that, relative to latest consistency-distilled dLLM of comparable scale, pcLLM occupies a more favorable point in the speed–quality trade-off space.

**Speculative decoding and recent dLLM baselines.**    Speculative decoding (SD) forms widely used family of AR acceleration methods. To place pcLLM among such approaches, this appendix includes comparisons against two recent SD methods, EAGLE-3 (Li et al., 2025) and HASS (Zhang et al., 2025), which represent stronger baselines than earlier methods such as Medusa and Medusa-2.

The comparison in Table 6 also includes two recent dLLM baselines, Fast-dLLM v2 (Wu et al., 2025a) and SDAR (Cheng et al., 2025), in addition to the community dLLM and D2F variants discussed in the main text. Fast-dLLM v2 improves blockwise diffusion efficiency via enhanced scheduling and caching, while SDAR introduces a synergistic diffusion–autoregressive paradigm for scalable sequence generation.

## B   MAPPING NOISE SCHEDULE TO TRAINING SEQUENCE FOR PROGRESSIVE CONSISTENCY DISTILLATION

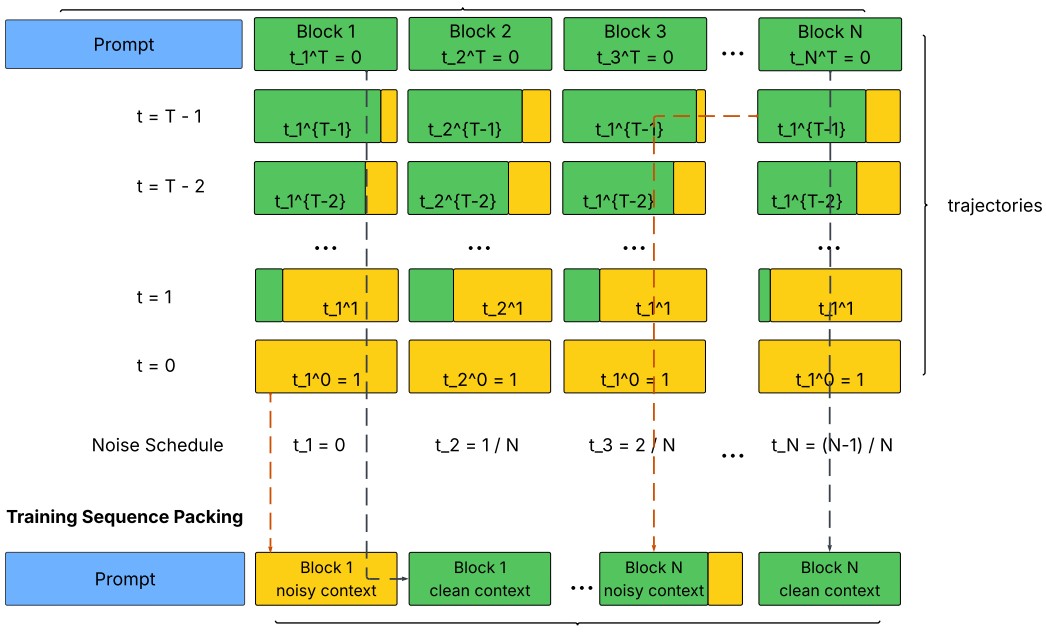

Figure 5: Illustration of the progressive noise schedule and training sequence packing. For each block $i$ over a total of $T_i$ decoding steps, we select the trajectory step whose fraction of unconverged tokens matches the scheduled noise ratio $t_i$ to form a noisy block (dashed red line), and pair it with the corresponding clean block (dashed dark line). The packed training sequence at the bottom interleaves all noisy and clean blocks, yielding $2N$ blocks so that a single forward pass can compute both AR and consistency losses.

We elaborate the process of mapping noise schedule to arrive at the training sequence in Figure 1.

For each training sample, let the target model's complete generation of length $L$ be $\boldsymbol{y}$. Given a training-time block size $n$ and a noise schedule $W$ (e.g., the linear progressive schedule in Eq. 2), we partition $\boldsymbol{y}$ into $N = \lceil L/n \rceil$ blocks of size $n$. The schedule $W$ is applied over a window of $w$ blocks, yielding noise ratios $t_i$ defined in Eq. 7. For each block, we select the point along its Jacobi trajectory whose fraction of unconverged tokens (number of unconverged tokens/$n$) is closest to $t_i$, and use that point to form the corresponding noisy block.

A complete training sequence contains both noisy and clean blocks. Clean blocks are the original partitions of $\boldsymbol{y}$, while noisy blocks are constructed as above. We interleave each noisy block with its corresponding clean block so that a single forward pass, together with the custom attention mask in Figure 2, produces teacher logits on clean blocks for the AR loss and student logits on noisy blocks for the consistency loss. Under the progressive noise schedule, the longest consecutive noisy span within any block is $O(\lceil tn \rceil)$, which is much smaller than the naive $O(nN)$ worst case where every token in every block is noisy.

## C   UNDERSTANDING TPF AND FLOPS TRADE-OFF

To estimate how many tokens can be decoded in parallel before hitting the hardware roofline, we profile generation-only latency as a function of the total number of simultaneously decoded tokens (horizontal axis in Figure 6), sweeping several block sizes $n_{\text{token\_seq\_len}}$. On H200 and B200 (left and

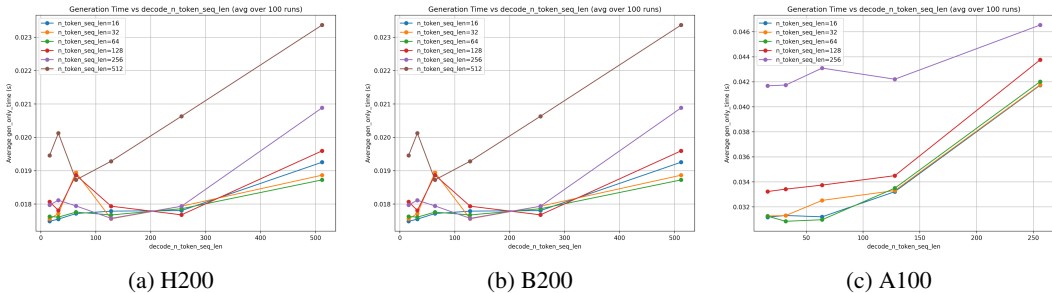

(a) H200            (b) B200            (c) A100

Figure 6: Generation-only latency versus total number of parallel decoded tokens across three hardware platforms (A100, H200, B200).

middle panels), the curves for $n_{\text{token\_seq\_len}} \in \{16, 32, 64, 128\}$ are essentially flat as we increase the parallel token count up to $\approx 256$ tokens, and only start to grow noticeably when we push beyond that to $512$ tokens. This plateau followed by an approximately linear region is the empirical roofline: up to $\sim 256$ batched tokens the GPU has spare FLOPs and KV bandwidth, so extra tokens are almost "free," whereas beyond that point the device becomes compute-/memory-bound and latency scales roughly linearly.

On A100 (right panel of Figure 6), the plateau is shorter: generation time is nearly constant up to $\sim 128$ parallel tokens, but increases steeply once we go beyond 128 and approaches linear scaling by 256 tokens. Taken together, these measurements suggest operating near the "knee" of each roofline, which corresponds to $\approx 128$ parallel tokens on A100 and $\approx 256$ parallel tokens on H200/B200. This motivates our final configuration: block size 64 with verification size 4 on H200 and B200 ($64 \times 4 = 256$ tokens), which maximizes FLOPs utilization without hurting wall-clock performance.

These roofline measurements imply a FLOPs budget on each GPU: once the parallel token count approaches the hardware knee, additional tokens incur an almost linear increase in cost. Consequently, there is an explicit TPF–FLOPs tradeoff: configurations with larger blocks and more aggressive parallelism achieve higher TPF, but the extra FLOPs consumption can saturate the hardware and even degrade wall-clock latency.

## D   Inference Configuration Search

Because of this TPF-FLOPs trade-off, choosing an inference configuration is no longer a matter of simply maximizing block size or verification depth: **the configuration must respect the FLOPs budget implied by the roofline of the target GPU**. Once $K = 2$ and $r = 0.85$ (initialization threshold) are fixed as training-optimal values from a separate grid search (as discussed in Section 4.3, the remaining degrees of freedom at inference are the block size $n_{\text{token\_seq\_len}}$ and the $n$-gram verification size, which jointly determine how much parallel draft/verify work is done per step under a given hardware constraint.

To explore this space, we perform a grid search over block sizes $n_{\text{token\_seq\_len}} \in \{8, 16, 32, 64, 128, 256\}$ and $n$-gram verification sizes $n_{\text{gram}} \in \{1, 2, 4, 8, 12\}$, measuring the achieved tokens per second for each pair on the target GPU. Since the raw grid is relatively coarse, we fit a smooth surface over the discrete measurements and use it as a surrogate for continuous hyperparameter selection. Specifically, we construct a 2D polynomial design matrix in (block size, $n$-gram size) of total degree up to 6, select the best degree by mean squared error, and then interpolate the fitted surface onto a dense grid using `scipy.interpolate.griddata` with a light Gaussian-like smoothing pass.

The results are shown in Figure 7, and the resulting surfaces reveal a clear optimum region: tokens-per-second peaks at moderate block sizes and medium $n$-gram verification, with the global maximum near $n_{\text{token\_seq\_len}} \approx 64$ and $n_{\text{gram}} \approx 4$. Very small blocks or $n$-gram verification size underutilize the available FLOPs, while very larger choices push the system closer to the roofline and begin to degrade wall-clock latency. This analysis justifies the final choice of using block size 64

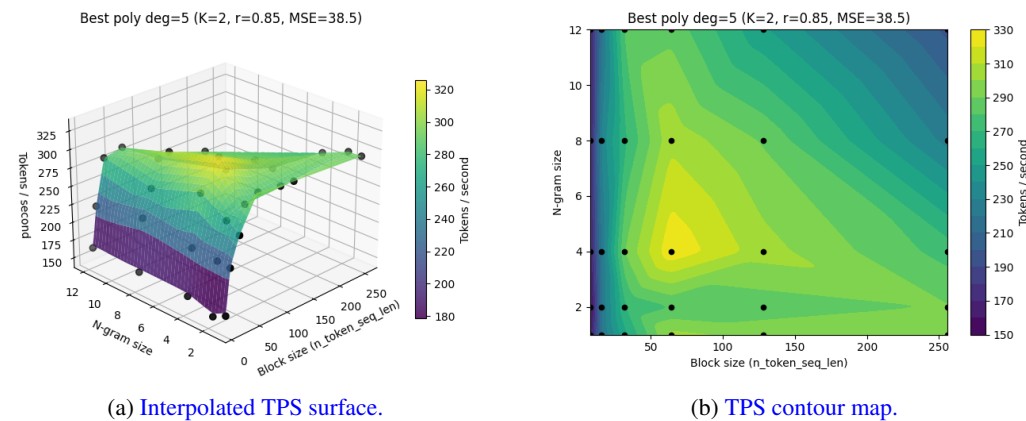

(a) Interpolated TPS surface.  (b) TPS contour map.

Figure 7: Tokens-per-second (TPS) as a function of block size $n_{\text{token\_seq\_len}}$ and $n$-gram verification size for $K = 2$ and $r = 0.85$. Black dots indicate measured configurations; the surface and contours are obtained by interpolation and light smoothing.

and $n$-gram size $4$ on H200/B200, which lies near the empirical optimum under each GPU's FLOPs budget.