# OpenReview forum: "AR Models can be Faster and More Accurate Parallel Decoders than Diffusion LLMs"
_ICLR.cc/2026/Conference — Submitted to ICLR 2026_

### Official Review · Reviewer_hCKy · 2025-10-27

**Soundness:** 1
**Presentation:** 2
**Contribution:** 2
**Rating:** 2
**Confidence:** 4

**Summary:**

This paper introduces pcLLM, a novel training method called progressive consistency distillation that trains autoregressive (AR) models to function as efficient parallel decoders. The key contribution lies in three components designed to improve on prior work like CLLM: a progressive noise schedule, a new sequence packing strategy, and a noise-aware causal attention mask. This progressive approach aims to solve a key challenge in consistency-based training: the difficulty of accurately predicting an entire block of tokens when conditioned on long, unconverged (noisy) preceding text. The authors report significant speedups on coding benchmarks (up to 3.8x vs. standard AR with 1-3% accuracy difference). Compared to the strong CLLM baseline, pcLLM is up to 44% faster, but with a mixed impact on quality: it shows a 3.2% absolute accuracy drop on HumanEval while simultaneously achieving a 2.0% absolute gain on MBPP.

**Strengths:**

- The paper is well-structured and clearly motivated. The authors do a commendable job of positioning their work against prior methods (like CLLM) and explaining the technical rationale for the new components.
- The progressive noise schedule and noise-aware causal masking are an interesting and notable evolution of the original CLLM approach, directly tackling the known challenge of training with large, noisy context blocks.
- From a systems perspective, the addition of sequence packing is a clever and practical optimization that contributes to the method's efficiency.
- The evaluation provides a thorough investigation of inference speedups relative to block size (as seen in Figure 4b). The authors demonstrate a clear, positive impact on speed (up to 3.8x vs. AR) and show their method is competitive against a strong baseline (CLLM). Notably, on the MBPP benchmark, the proposed pcLLM (MR) method is both faster (+27%) and more accurate (+2.0%) than the CLLM baseline.
- The authors include a clear and effective ablation study (Table 4a) that successfully isolates the benefit of their proposed noise-conditioned mask, validating its impact on performance over other masking strategies.
- The techniques presented, particularly the progressive distillation schedule, are promising and could be built upon by future work in Jacobian-based parallel decoding.

**Weaknesses:**

- W1. A major weakness is the failure to compare against speculative decoding (SD). The authors' justification for omitting this key baseline is unconvincing. This comparison is critical because SD is a lossless method, guaranteeing output identical to the original AR model. In contrast, pcLLM is lossy, as its re-training process incurs a 3.2% absolute accuracy drop on HumanEval relative to that same original AR model. Without this baseline, the practical benefit of the paper's complex, lossy method over a simpler, lossless alternative is unknown.
- W2. The main results in Table 1 are difficult to interpret, as the block size k used for the pcLLM and CLLM methods is not specified. The paper's own ablation studies show that k is a highly sensitive parameter that significantly impacts both speedup and accuracy. This omission is problematic because it is unclear if the 3.62x speedup on HumanEval and the 84.8 accuracy score were achieved at the same k. This makes it difficult for the reader to assess the method's true speed-versus-accuracy trade-off from the main results table.
- W3. The claim that the method "largely" preserves performance is an overstatement. A 3.2% absolute drop on HumanEval (88.0% → 84.8%) is a very significant degradation in quality. Furthermore, the results are mixed, not a clear win (it wins on MBPP but loses badly on HumanEval), and the evaluation is limited to a single domain (coding), making it hard to generalize the findings to other tasks.
- W4. The evaluation is missing several key analyses. First, there is no study on the effect of model size. It is unclear if the reported speedups will hold for larger, more practical models (e.g., 70B, 175B), which is a key consideration for the method's impact. Second, the paper fails to provide Pareto curves (speed vs. accuracy) for both pcLLM and CLLM. A full curve, generated by sweeping the block size k, is necessary for a fair, holistic comparison. The current tables only present a single operating point.
- W5. The method's complexity, arising from both its multi-objective training and iterative inference engine, introduces overhead. This is confirmed by the ablation study (Fig 4b), which shows the method is slower than standard AR at small block sizes and hits a hard speedup ceiling. This high overhead and limited scalability call the method's practical utility into question.

**Questions:**

- Could the authors provide a direct comparison (wall-clock speedup vs. accuracy) of pcLLM against a strong, lossless SD baseline to properly situate the paper's contribution?
- What block size $k$ was used to generate the main results in Table 1? To better understand the trade-offs, could you show both speedup and HumanEval accuracy for pcLLM at various block sizes (e.g., k=8, 16, and 50)?
- How do the reported speedups and quality trade-offs for pcLLM scale to larger, more practical models (e.g., 13B or 70B)?
- To provide a fairer, more holistic comparison, would it be possible for the authors to provide speed-vs-accuracy Pareto curves for both pcLLM and CLLM, generated by sweeping the block size $k$? This would be more informative than the single data point in Table 1.

---

> ### Author Response · Authors · 2025-11-29
>
> ### **W1 & Q1: Could the authors provide a direct comparison (wall-clock speedup vs. accuracy) of pcLLM against a strong, lossless SD baseline to properly situate the paper's contribution?**
>
> We agree that speculative decoding baselines are highly relevant families of AR acceleration methods, and we have added comparison with two latest SD baseline (EAGLE-3 [1] and HASS [2], which are stronger than Medusa, and Medusa-2).
>
> We also add further comparison with latest dLLM baselines (Fast-dLLM v2 [3], SDAR [4]), as well as the consistency distilled dLLM baseline dParallel (as discussed in response to W1/Q1).
>
> | Family          | Method        | Acc. $\uparrow$ | TPF $\uparrow$     | TPS $\uparrow$        | Speedup vs. AR $\uparrow$ |
> |-----------------|----------------------|---------------|------------------|--------------------------|-------------------------|
> | AR baseline    | AR (greedy)   | 87.8           | $1.00$           | $83$               |   $1.00\times$         |
> | AR + Jacobi        | Jacobi         |   87.8        |  $1.05$         |     $84.7$       |     $1.02\times$        |
> | dLLM            | Fast-dLLM v2    | 63.4          |  $1.00$         |  $83.29$          |    $1.00\times$        |
> | dLLM            | SDAR                | 78.7          |  $2.36$            | $31.46$               | $0.38\times$     |
> | dLLM (distilled) | dParallel       | 54.3          |    $2.90$      |  $175.15$                 | $2.11\times$      |
> | AR + Spec-Dec   | EAGLE-3*    | 68.9*        | $6.38$            | $246.10$          |    $2.97\times$          |
> | AR + Spec-Dec   | HASS*         |  61.6*             | $5.53$       | $280.29$           |   $3.37\times$     |
> | AR + Spec-Dec    | CLLM          | 87.8               | $2.80$        |         $207.4$         | $2.50\times$    |
> | AR + Jacobi         | pcLLM             |     84.8       |     $3.96$     |      $299.5$         |  $3.61\times$       |
> | AR + Jacobi        | pcLLM (MR)     |      84.8      |      $4.21$     |      $319.57$     |    $3.85\times$   |
>
> For Eagle-3 and HASS, we use the best available checkpoint provided by the authors for speed measurement. For Eagle-3, we use DeepSeek-R1-Distill-Llama-8B as the base model; for HASS, we use LLaMA3-Instruct as the base model.
>
> From the table, we can observe that currently pcLLM achieves both the best TPF as well as a wall-clock speedup at the cost of small performance degradation from the original AR model baseline.
>
> [1] Li, Yuhui, et al. “EAGLE-3: Scaling up Inference Acceleration of Large Language Models via Training-Time Test.” arXiv, 3 Mar. 2025, arxiv.org/abs/2503.01840.
>
> [2] Zhang, Lefan, et al. “Learning Harmonized Representations for Speculative Sampling.” arXiv, 28 Aug. 2024, arxiv.org/abs/2408.15766.
>
> [3] Wu, Chengyue, et al. “Fast-dLLM v2: Efficient Block-Diffusion LLM.” arXiv, 30 Sept. 2025, arxiv.org/abs/2509.26328.
>
> [4] Cheng, Shuang, et al. “SDAR: A Synergistic Diffusion-AutoRegression Paradigm for Scalable Sequence Generation.” arXiv, 7 Oct. 2025, arxiv.org/abs/2510.06303
>
>
> ### **W2 & Q2: What block size k was used to generate the main results in Table 1? To better understand the trade-offs, could you show both speedup and HumanEval accuracy for pcLLM at various block sizes (e.g., k=8, 16, and 50)?**
>
> our experiments use greedy decoding, so Jacobi decoding is **distribution-equivalent to standard AR**: all tokens are accepted causally and only the top-1 logit is used for verification. As a result, there is no quality trade-off across block sizes, the generation is identical AR, regardless of the chosen block size.
>
> What does depend on block size is the degree of parallelism and therefore total FLOPs consumption: larger blocks increase TPF (tokens per forward) by drafting more tokens in parallel but also raise per-step FLOPs. The relevant trade-off is therefore TPF vs. FLOPs, i.e. **finding the block size that maximizes TPS on a given hardware budget**. We perform this analysis on the B200 machine and report the resulting TPS curves under different block size choices at k=8, 16, 32, 64, 128, and 256.
>
> In the paper, we use block size k=128 to generate the main results in Table 1 (this is mentioned in Section 4.2 of the paper), except for pcLLM (MR) since MR takes extra FLOPs for multiblock decoding and parallel verification, where we use k=64 (according to the inference configuration grid search results).
>
> We have also provided further illustrations and details discussion about TPF vs. FLOPs tradeoff in Appendix D.

---

> ### Author Response · Authors · 2025-11-29
>
> ### **W3: The claim that the method "largely" preserves performance is an overstatement.... the results are mixed, not a clear win (it wins on MBPP but loses badly on HumanEval), and the evaluation is limited to a single domain (coding), making it hard to generalize the findings to other tasks.**
>
> We agree that using the word “largely” without qualification can be misleading and will soften this phrasing. Our goal is to trade a small variation in accuracy for a substantial, practical speedup. On HumanEval, pcLLM (MR) incurs a 3.2% performance drop while still delivering nearly 4x wall-clock speedup (and 4.2x higher TPF than AR). We view this as a meaningful operating point for latency-sensitive coding applications.
>
> To test generality beyond code and short CoT, we further applied the same pcLLM training recipe to Qwen2.5-Math-7B-Instruct on GSM8K and MATH, using longer trajectories (up to 1k tokens). On GSM8K we obtain 91.4% (–1.0% from AR) at 3.71x TPS, and on MATH we obtain 77.4% (+0.4% from AR) at 3.68x TPS. The small accuracy fluctuation on MATH (a slight gain rather than a loss) suggests that the variance introduced by pcLLM training is within a reasonable range across benchmarks. The results are summarized in the table below:
>
>
> | Dataset | TPF $\uparrow$  | Accuracy $\uparrow$   |
> |---------|------|----------------|
> | GSM8k   | 4.04 | 91.4 (-1.0)    |
> | MATH    | 3.98 | 77.4 (+0.4)    |
>
> ### **W4 & Q3: How do the reported speedups and quality trade-offs for pcLLM scale to larger, more practical models (e.g., 13B or 70B)?**
>
> Our experiments are conducted on 7B-scale models due to compute constraints, so we do not claim empirical results for 13B/70B at this stage. However, please also note progressive consistency distillation does not rely on size-specific architectural tricks, so we expect the method will transfer to larger models.
>
> ### **W5: The method's complexity, arising from both its multi-objective training and iterative inference engine, introduces overhead... This high overhead and limited scalability call the method's practical utility into question.**
>
> **We want to kindly point out that the interpretation that our method is slower than AR at small block sizes is incorrect**. Figure 4b shows a consistent speedup across all tested block sizes: even at block size 8, our method achieves TPF > 2, i.e., more than 2 tokens per forward pass, already outperforming standard AR.
>
> The training overhead is a one-time overhead incurred before inference time, and at inference cost incurred from extra computation when employing our technique with multi-block Jacobi decoding with rejection recycling is more than amortized by the increased TPF, especially considering idle FLOPs from latest generations of GPUs such as NVIDIA B200. We will clarify this in the text and explicitly highlight that **our approach is never slower than AR** as shown in Figure 4b.
>
> ### **Q6: To provide a fairer, more holistic comparison, would it be possible for the authors to provide speed-vs-accuracy Pareto curves for both pcLLM and CLLM, generated by sweeping the block size k? This would be more informative than the single data point in Table 1.**
>
> We thank reviewers for the suggestion. Instead of a single operating point, we sweep the block size k=16,32,64,128,256 for both CLLM and pcLLM and report the TPS speedup over AR in the table below. We use greedy decoding, so Jacobi decoding is distribution-equivalent to AR (all tokens are accepted causally with top-1 verification), meaning accuracy is effectively unchanged across block sizes. And **the most relevant trade-off is therefore TPF vs. FLOPs at different block sizes (to achieve highest wall-clock speedup TPS)**:
>
> | Method | Block size = 16 |   32     |    64     |    128     |    256     |    Acc    |
> |------------|-------------------|------------|------------|-------------|-------------|-----------|
> | CLLM |  $2.4\times$    | $2.5\times$ | $2.5\times$ | $2.4\times$ | $2.0\times$ | 87.8 |
> | pcLLM | $2.7\times$ | $3.5\times$ | $3.9\times$ | $3.7\times$ | $3.3\times$ | 84.8 |
>
> Entries are speedup in TPS vs. AR. pcLLM values are profiled using multiblock decoding with rejection recycling with the optimal choice of inference configuration at K=2, r=0.85, ngram verification size=4.
>
> From the table, we can see pcLLM consistently provides higher speedups than CLLM at the same block size, especially at k=32, 64. Scaling block size to 256 reduces wall-clock speedup since it requires more FLOPs for each forward pass (despite potential improvement in TPF).

---

### Official Review · Reviewer_TYFF · 2025-10-28

**Soundness:** 1
**Presentation:** 1
**Contribution:** 2
**Rating:** 0
**Confidence:** 4

**Summary:**

The paper introduces pcLLM, a training and inference framework that aims to turn autoregressive (AR) LLMs into parallel decoders via "progressive consistency distillation", coupled with Jacobi-style iterative decoding and inference-time optimizations (rejection recycling). Empirically, the paper reports 3.6-4.0x wall-clock speedups on coding benchmarks with modest accuracy drops and argues advantages over diffusion-based LLMs, and argues that the training curriculum is more suited than the original CLLMs training procedure, as learning to predict shorter blocks first is easier and more stable. No comparisons with speculative decoding are included.

**Strengths:**

**Significance**: the pcLLMs are faster than vanilla CLLMs, and current diffusion LLMs.

**Weaknesses:**

## Summary of the weaknesses
- The submission is difficult to read. I would not be able to implement the method, as multiple details are unclear or not discussed. There is no appendix in the submission, which would have been a good place to clarify the method.
- Comparing pcLLMs to diffusion LLMs only, and not speculative decoding, a conscious choice of the authors (line 351), seems unjustified.


## Detailed weaknesses
1. Missing citations in the introduction for important milestones in discrete diffusion models (D3PM, SEDD, MDLM, MD4, RADD)
2. It is unclear how the noise ratios $t_i$, introduced on line 184, are used. Given trajectories generated from the original cLLM, it seems the authors want to pick points along the consistency trajectory, such that each block has a certain noise level, close to the ratios $t_i$s. However, there is no guarantee that trajectories with the desired ratios exist. An alternative explanation, is that the authors *combine* different trajectories, by extracting blocks from different trajectories, and combining them into a training example. Can you clarify how the training examples are selected, in particular, such that each block has a desired ratio of noisy tokens?
3. The sentence packing strategy is unclear. There are no explanations on the mechanism, simply a reference to the figure (lines 229, 230), which does not suffice to explain it.
4. Experiments (line 233): "*Collecting additional rounds of Jacobi trajectories from intermediate checkpoints help*". This is not an adequate amount of details. What fraction of trajectories are sampled from the intermediate checkpoints? How frequently? Do you replace examples from the original cLLM when introducing the trajectories from the intermediate checkpoints? These are important questions that are not discussed at all.
5. Experiments (line 235): You argue that the block sizes are increased progressively during training. How progressively? From line 316, it seems you double the block size from 16 to 32, however it is not clear at which point of training. Are you doubling the block size after half the total number of steps?
6. **Missing important baselines** (Line 351): The decision to exclude speculative decoding from the comparison appears unjustified. Including it would have been more relevant than comparing against dLLMs, which are currently known to underperform autoregressive models in both performance and speed, particularly when KV-caching optimizations are not applied. Moreover, omitting comparisons with multi-token prediction methods such as Medusa also seems questionable. Your approach requires fine-tuning the model, and although Medusa introduces additional decoding heads, both methods aim to accelerate sampling from autoregressive models. Therefore, including at least one speculative decoding baseline, as well as Medusa, would make the evaluation more complete and fair.
7. It is not clear what the trade-off is between speed-up and performance on coding benchmark is, when using different block sizes during sampling.

**Questions:**

1. You introduce a distillation method for consistency LLMs, yet compare pcLLMs with *un-distilled* diffusion models. A fairer comparison would be between pcLLMs, and distilled diffusion LLMs, e.g. with SDTT [1], or Di4C [2]. Is there a reason why you did not compare with distilled dLLMs?
2. Can you clarify how the training examples are selected, in particular, such that each block has a desired ratio of noisy tokens (cf weakness 2)
3. What is the window size $w$? How does it compare to $n$? From line 183, it seems that $n$ already represented the length of the smaller blocks, does that mean that $n=w$ ?
4. What is $t$, on line 192? Is the the value of $t_i$ in an arbitrary block (so that the argument holds for any block)?
5. *Progressive distillation loss* (line 194): In your work, it seems you are using noisy blocks in the prefix. This is surprising, since cLLMs sample new blocks based on fully denoised blocks. In your case, you start generating future blocks before the previous ones are fully denoised, is that correct? It is surprising that such approach would be faster than generating blocks one-by-one, as once a block has converged, one can use KV-caching, hence lookup will be faster
6. You mention CoT generation for reasoning model (last line of page 4). In Figure 4a (caption), you argue that the timings are performed with a prompt of length 128, and generating 256 tokens. This is rather short. Did you measure the efficiency of your approach for longer generation, e.g. generating 2048, 4096 tokens?



[1]: Beyond Autoregression: Fast LLMs via Self-Distillation Through Time, Deschenaux and Gulcehre

[2]: Distillation of Discrete Diffusion through Dimensional Correlations, Hayakawa et al.

---

> ### Author Response · Authors · 2025-11-28
>
> ### **W1: Missing citations in the introduction for important milestones in discrete diffusion models (D3PM, SEDD, MDLM, MD4, RADD)**
>
> ### **Q1: You introduce a distillation method for consistency LLMs, yet compare pcLLMs with un-distilled diffusion models. A fairer comparison would be between pcLLMs, and distilled diffusion LLMs, e.g. with SDTT [1], or Di4C [2]. Is there a reason why you did not compare with distilled dLLMs?**
>
> We thank the reviewers for bringing up important milestones in discrete diffusion models that lead to recent advancements of diffusion large language models. We have added the citations to the pioneering work in the paper.
>
> We acknowledge that a distilled dLLM baseline would be an important baseline to map where pcLLM is in comparison with current dLLM training techniques. Unfortunately, since both SDTT and Di4C are much smaller pretrained discrete diffusion models in comparison with Qwen-2.5-7B, we instead compare pcLLM with dParallel [1], a consistency distilled dLLM using the model’s trajectory.
>
> The results are summarized in Table R2 along with other speculative decoding baselines. It can be seen that a noticeable gap remains between dParallel and pcLLM, where pcLLM outperforms dParallel by 8% on GSM8K and 28% on HumanEval with 20% and 80% higher TPF (and TPS) respectively.
>
> [1] Chen, Zigeng, et al. “dParallel: Learnable Parallel Decoding for dLLMs.” arXiv, 30 Sept. 2025. arXiv.org, https://arxiv.org/abs/2509.26488.

---

> ### Author Response · Authors · 2025-11-28
>
> ## Mapping Noise Schedule to Training Sequence
> ### **W2: It is unclear how the noise ratios t_i, introduced on line 184, are used ... Can you clarify how the training examples are selected, in particular, such that each block has a desired ratio of noisy tokens?**
>
> ### **Q2: Can you clarify how the training examples are selected, in particular, such that each block has a desired ratio of noisy tokens (cf weakness 2)**
>
> ### **Q3: What is the window size w? How does it compare to n? From line 183, it seems that n already represented the length of the smaller blocks, does that mean that  n=w?**
>
> You can refer to Appendix B and we have added an illustration to clarify the training sequence packing process and how it maps noise schedule to the sequence.
>
> We want to clarify our noise schedule and its relationship to the training sequence as follows. For each training sample, let the target model’s complete generation of length $L$ be $\mathbf{y}$. Given a training-time block size $n$ and a noise schedule $W$ (e.g., the linear progressive schedule in Eq. 2), we partition $\mathbf{y}$ into $N = \lceil L / n \rceil$ blocks of size $n$ (see Eq. 4 from the paper). To assign noise ratios to these blocks, we must specify how many incremental steps of $W$ will be used: that is, the total number of blocks over which the noise ratios $t_i$ from $W$ are consumed. We denote this number of steps as $w$, the window-size parameter introduced in Eq. 7.
>
> We then extract blocks from different sequence trajectory (for all $N$ blocks) by identifying the sequence (a point on the trajectory) with the closest rounding error when comparing (# unconverged tokens / n) with the corresponding noise ratio $t_i$.
>
> ### **W3: The sentence packing strategy is unclear. There are no explanations on the mechanism, simply a reference to the figure (lines 229, 230), which does not suffice to explain it.**
>
> In the above response, we provide details on how we map noise ratios to all blocks from a training sequence (and form noisy blocks). However, as illustrated in Figure 2, a complete training sequence composes both **noisy blocks** and **clean blocks**. The clean blocks are simply partitions blocks from the target model’s complete generation. To construct a complete training sequence, we simply interleave each noisy block with its corresponding clean block. They blocks are packed this way such that we can calculate both loss in Eq. 9 in one forward pass (with the customized attention mask in Figure 2, all clean blocks yield teacher logits conditioned on all clean blocks for AR loss computation and noisy blocks yield student logits needed for consistency loss computation).
>
> ### **Q4 What is t, on line 192? Is the value of t_i in an arbitrary block (so that the argument holds for any block)?**
>
> Yes, it holds for $t_i$ in any arbitrary block, as $\lceil tn \rceil$ is the consecutive noisy token count using our progressive noise schedule.
>
> Without applying the progressive noise schedule, since all noisy blocks are attended to noisy blocks in noisy-context conditioned training, consider the worst case scenario where **all noisy blocks contain ONLY noisy unconverged tokens**, the longest span is $O(nN)$, where $N$ is the total number of blocks. Since $t \leq 1, n << L$, the worst case noisy token span $O(\lceil tn \rceil)$ with progressive noise schedule is therefore much better.
>
> In the paper we write $O(\lceil tnw \rceil)$ with the assumption $t=1 (\text{all tokens are noisy across all blocks}), w=N$ with a homogenous linear progressive noise schedule from 0 to 1 applied to all blocks. We apologize for the confusion and have revised our manuscript to clarify.
>
> For more implementation on data preparation, we have also attached our data processing code in the supplementary materials for your reference. Feel free to check it out and we will be more than happy to provide further clarifications.

---

> > ### Author Response · Authors · 2025-11-28
> >
> > ## Training Details
> >
> > ### **W4: Experiments (line 233): "Collecting additional rounds of Jacobi trajectories from intermediate checkpoints help". This is not an adequate amount of details. What fraction of trajectories are sampled from the intermediate checkpoints? How frequently? Do you replace examples from the original cLLM when introducing the trajectories from the intermediate checkpoints? These are important questions that are not discussed at all.**
> >
> > On OpenCodeInstruct, we use 400k trajectories (the same set of trajectories from the first round) collected to launch the additional round of training, but with a **larger block size at 32** (in comparison to using a block size of 16 from round 1). This round of training only starts after the training is complete from the first round with 450k training examples. CLLM uses different training data than ours (CLLM trains the models on GSM8K training set for math, Spider training set for text2SQL, CodeSearchNet for Python coding).
> >
> > ### **Q5: Progressive distillation loss (line 194): In your work, it seems you are using noisy blocks in the prefix. This is surprising, since cLLMs sample new blocks based on fully denoised blocks. In your case, you start generating future blocks before the previous ones are fully denoised, is that correct? It is surprising that such approach would be faster than generating blocks one-by-one, as once a block has converged, one can use KV-caching, hence lookup will be faster.**
> >
> > We want to clarify that the progressive distillation loss with noisy blocks is **only used during training**: we attend noisy blocks to noisy blocks with progressive noise so the model learns to make good predictions even when earlier tokens are not fully denoised.
> >
> > And yes, at inference, when applying Jacobi decoding (the row without MR in Table 1) we still decode with fully denoised prefixes and standard KV caching.
> >
> > **multi-block decoding simply uses otherwise idle FLOPs to draft future blocks in parallel while the current block is converging, and those future tokens are later verified with Jacobi decoding before being committed**. As a result, the final outputs and KV cache are identical to block-by-block decoding, but we need fewer global iterations, which yields a net speedup rather than a slowdown.
> >
> > ### **W5: Experiments (line 235): You argue that the block sizes are increased progressively during training. How progressively? From line 316, it seems you double the block size from 16 to 32, however it is not clear at which point of training. Are you doubling the block size after half the total number of steps?**
> >
> > Following response to W4, in the second round of training, we choose the combination n=32, w=16 by conducting grid search through possible n=32 combinations. In the rebuttal we provide a more comprehensive grid-search result for the optimal configuration using both block size n=32 and n=64 at varying window sizes as shown in Table R1.
> >
> > | n   | Window | 25k steps (Acc / TPF) | 50k steps (Acc / TPF) | 75k steps (Acc / TPF) | 100k steps (Acc / TPF) |
> > |-----|--------|-------------------------|--------------------------|--------------------------|---------------------------|
> > | 32  | w32    | 86.0 / 3.17             | 85.4 / 3.42              | 85.4 / 3.46              | 83.5 / 3.62               |
> > | 32  | w16    | 86.0 / 3.40             | 86.0 / 3.70              | 85.4 / 3.97              | **84.8 / 4.18**              |
> > | 32  | w8     | 84.8 / 3.48             | 84.8 / 3.76              | 83.5 / 4.02              | 82.9 / 4.26               |
> > | 64  | w64    | 85.4 / 3.21             | 85.4 / 3.29              | 84.8 / 3.46              | 83.5 / 3.57               |
> > | 64  | w32    | 85.4 / 3.51             | 84.8 / 3.77              | 83.5 / 4.18              | **83.5 / 4.44**               |
> > | 64  | w16    | 86.0 / 3.47             | 85.4 / 3.83              | 85.4 / 4.08              | 82.9 / 4.52               |
> >
> > **Table R1: ablation study on training hyperparameter choice for an additional round of progressive consistency distillation. Maximum TPF is measured by applying grid search to inference configurations.**
> >
> > In Table R1, a few observations can be made:
> > - As the block size scales, larger window size becomes a suboptimal choice. And adding noise too aggressively (with the respect to block size) will result in training instability.
> > - As training data scales, maximum TPS will saturate among correctly generated samples.

---

> ### Author Response · Authors · 2025-11-29
>
> ### **W6: Missing important baselines (Line 351): The decision to exclude speculative decoding from the comparison appears unjustified... Therefore, including at least one speculative decoding baseline, as well as Medusa, would make the evaluation more complete and fair.**
>
> We agree that speculative decoding baselines are highly relevant families of AR acceleration methods, and we have added comparison with two latest SD baseline (EAGLE-3 [1] and HASS [2], which are stronger than Medusa, and Medusa-2).
>
> We also add further comparison with latest dLLM baselines (Fast-dLLM v2 [3], SDAR [4]), as well as the consistency distilled dLLM baseline dParallel (as discussed in response to W1/Q1). TPS is measured on a single B200 GPU and TPF is hardware independent.
>
> | Family          | Method        | Acc. $\uparrow$ | TPF $\uparrow$     | TPS $\uparrow$        | Speedup vs. AR $\uparrow$ |
> |-----------------|----------------------|---------------|------------------|--------------------------|-------------------------|
> | AR baseline    | AR (greedy)   | 87.8           | $1.00$           | $83.0$               |   $1.00\times$         |
> | AR + Jacobi        | Jacobi         |   87.8        |  $1.05$         |     $84.7$       |     $1.02\times$        |
> | dLLM            | Fast-dLLM v2    | 63.4          |  $1.00$         |  $83.3$          |    $1.00\times$        |
> | dLLM            | SDAR                | 78.7          |  $2.36$            | $31.5$               | $0.38\times$     |
> | dLLM (distilled) | dParallel       | 54.3          |    $2.90$      |  $175.2$                 | $2.11\times$      |
> | AR + Spec-Dec   | EAGLE-3*    | 68.9*        | $6.38$            | $246.1$          |    $2.97\times$      |
> | AR + Spec-Dec   | HASS*         |  61.6*             | $5.53$       | $280.3$           |   $3.37\times$     |
> | AR + Spec-Dec    | CLLM          | 87.8               | $2.80$        |         $207.4$         | $2.50\times$    |
> | AR + Jacobi         | pcLLM             |     84.8       |     $3.96$     |      $299.5$         |  $3.61\times$       |
> | AR + Jacobi        | pcLLM (MR)     |      84.8      |      $4.21$     |      $\textbf{319.6}$    |       $\mathbf {3.85\times}$  |
>
> For Eagle-3 and HASS, we use the best available checkpoint provided by the authors for speed measurement. For Eagle-3, we use DeepSeek-R1-Distill-Llama-8B as the base model; for HASS, we use LLaMA3-Instruct as the base model.
>
> From the table, we can observe that currently pcLLM achieves both the best TPF as well as a wall-clock speedup at the cost of small performance degradation from the original AR model baseline.
>
> [1] Li, Yuhui, et al. “EAGLE-3: Scaling up Inference Acceleration of Large Language Models via Training-Time Test.” arXiv, 3 Mar. 2025, arxiv.org/abs/2503.01840.
>
> [2] Zhang, Lefan, et al. “Learning Harmonized Representations for Speculative Sampling.” arXiv, 28 Aug. 2024, arxiv.org/abs/2408.15766.
>
> [3] Wu, Chengyue, et al. “Fast-dLLM v2: Efficient Block-Diffusion LLM.” arXiv, 30 Sept. 2025, arxiv.org/abs/2509.26328.
>
> [4] Cheng, Shuang, et al. “SDAR: A Synergistic Diffusion-AutoRegression Paradigm for Scalable Sequence Generation.” arXiv, 7 Oct. 2025, arxiv.org/abs/2510.06303.

---

> > ### Author Response · Authors · 2025-11-29
> >
> > ### **W7: It is not clear what the trade-off is between speed-up and performance on coding benchmark is, when using different block sizes during sampling.**
> >
> > Our coding experiments use greedy decoding, so Jacobi decoding is **distribution-equivalent to standard AR**: all tokens are accepted causally and only the top-1 logit is used for verification. As a result, there is no quality trade-off across block sizes, the generation is identical AR, regardless of the chosen block size.
> >
> > What does depend on block size is the degree of parallelism and therefore total FLOPs consumption: larger blocks increase TPF (tokens per forward) by drafting more tokens in parallel but also raise per-step FLOPs. The relevant trade-off is therefore TPF vs. FLOPs, i.e. **finding the block size that maximizes TPS on a given hardware budget**. We perform this analysis and report the resulting TPS curves under different block size choices at 8, 16, 32, 64, 128, and 256. We have elaborated more details on the TPF vs. FLOPs tradeoff in Section 4.3 of our paper.
> >
> > We have also provided further illustrations and details discussion about TPF vs. FLOPs tradeoff in Appendix D.
> >
> > ### **Q6: You mention CoT generation for the reasoning model (last line of page 4). In Figure 4a (caption), you argue that the timings are performed with a prompt of length 128, and generating 256 tokens. This is rather short. Did you measure the efficiency of your approach for longer generation, e.g. generating 2048, 4096 tokens?**
> >
> > To address the reviewer’s concern on reasoning tasks with longer context, we added additional experiment to train a pcLLM with longer generations (up to 1k tokens for the GSM8K and MATH datasets) using Qwen2.5-Math-7B-Instruct with the prompt set in OpenThoughts to generate trajectories. We apply pcLLM training under the same settings as used for the code tasks. The results compared to the original AR model are as follows:
> >
> > | Dataset | TPF $\uparrow$  | Accuracy $\uparrow$   |
> > |---------|------|----------------|
> > | GSM8k   | 4.04 | 91.4 (-1.0)    |
> > | MATH    | 3.98 | 77.4 (+0.4)    |
> >
> > We observed it maintains similar relative speedups over the AR baseline. We will include these extended-length throughput results in the revised appendix and clarify this scope in the main text.

---

### Official Review · Reviewer_XcmS · 2025-11-01

**Soundness:** 3
**Presentation:** 3
**Contribution:** 3
**Rating:** 4
**Confidence:** 3

**Summary:**

This paper proposes pcLLM, a progressive consistency distillation approach that converts autoregressive models into more efficient parallel decoders built upon Jacobi decoding. The method introduces a progressive noise schedule during consistency distillation. At inference time, the paper proposes rejection recycling and multi-block decoding to reuse stable n-gram segments and refine multiple blocks in parallel. The approach achieves up to ~4× speedup while aiming to preserve generation quality.

**Strengths:**

The progression from preliminaries to the progressive consistency distillation method is smooth, and the intuition behind the progressive noise schedule is clearly explained.

Figures and qualitative examples are effective, and several ablations (e.g., block size) help justify design decisions.

**Weaknesses:**

The explanation of multi-block decoding and rejection recycling is hard to follow, especially in Algorithm 1. While Figure 3 illustrate the behavior, a clearer algorithm description would improve clarity.

Evaluation is limited to HumanEval and MBPP, restricting generalization to broader reasoning or open-ended text generation tasks.

Section 4.2 claims that pcLLM “consistently achieves better accuracy,” but on HumanEval pcLLM (84.8) is lower than AR and CLLM baselines (~87–88).

Limited ablation on inference hyperparameters (e.g., spawn ratio r and number of active blocks K) makes the sensitivity of the decoding strategy unclear.

Noise schedule ablations are minimal. The reverse progressive schedule mainly demonstrates failure; additional variations (e.g., different window sizes or non-linear schedules) could better support the choice of the linear progressive cyclic schedule.

The paper does not explicitly discuss limitations or potential failure cases.

**Questions:**

Could the authors provide deeper intuition for why the linear progressive cyclic noise schedule is particularly effective?  Does such noise scheduling methodology possess any conceptual relations to diffusion-style denoising?

Beyond the example in Figure 2, can you provide more direct quantitative evidence isolating the contributions of multi-block decoding and rejection recycling to the speedup?

---

> ### Author Response · Authors · 2025-11-28
>
> ### **W1: Clarification on multiblock decoding settings.**
>
> Thank you for your suggestions. We have elaborate details on the inference algorithm and hyper-parameter choice in Section 4.2 and Section 4.3.
>
> ### **W2: Evaluation is limited to HumanEval and MBPP, restricting generalization to broader reasoning or open-ended text generation tasks.**
>
> Thank you for the suggestion. We have added additional experiments on math tasks, which is natural language generation. Specifically, we initialize the model with Qwen2.5-Math-7B-Instruct, use math problems as the prompt set in OpenThoughts to generate trajectories, and then apply pcLLM training under the same settings as used for the code tasks. The numbers in parentheses indicate the accuracy change compared to the original AR model, and **TPF denotes the number of tokens produced per forward pass**. Additional throughput and speedup results on A100 GPUs can be found in Table 3 of the updated manuscript.
>
> | Dataset | TPF $\uparrow$  | Accuracy $\uparrow$   |
> |---------|------|----------------|
> | GSM8k   | 4.04 | 91.4 (-1.0)    |
> | MATH    | 3.98 | 77.4 (+0.4)    |
>
> ### **W3: Section 4.2 claims that pcLLM “consistently achieves better accuracy,” but on HumanEval pcLLM (84.8) is lower than AR and CLLM baselines (~87–88).**
>
> Thank you for pointing this out. We have revised our manuscript to more precisely capture experiment results in Section 4.
>
> ### **W4: Limited ablation on inference hyperparameters (e.g., spawn ratio r and number of active blocks K) makes the sensitivity of the decoding strategy unclear.**
>
>
> We evaluate \sysname (MR) using Jacobi decoding on A100, H200, and B200 GPUs, sweeping block sizes up to $256$ (no latency penalty up to $64$ and only minor degradation at $128$ on H200, as shown from the profiling result in Figure 4a) and fixing accepted tokens to $2,3,4,5$.
>
> For the final inference setup, we use block size $64$, verification size $4$, such that tokens decoded in parallel each step is kept below $256$ tokens, with the initialization threshold fixed at $r=0.85$ (according to more detailed roofline profiling results provided in Appendix C).
>
> We have added a more detailed discussion to Section 4.3 and Appendix C.
>
> You can also find our modeling and profiling scripts in the supplementary materials for reference. Feel free to check it out and we will be more than happy to provide further clarifications.
>
>
> ### **W5: Noise schedule ablations are minimal. The reverse progressive schedule mainly demonstrates failure; additional variations (e.g., different window sizes or non-linear schedules) could better support the choice of the linear progressive cyclic schedule.**
>
>
> Thank you for the suggestion. We have added additional experiments using a block size of 64 with more window-size settings and non-linear noise schedules (e.g., quadratic progressive and random). For the quadratic progressive schedule, we divide the 0%–100% noise range into windows following a quadratic progression; for the random schedule, we uniformly sample noise levels from 0%–100%. The results of accuracy and TPF on HumanEval are shown in the table below. As observed, **the quadratic progressive scheduler is slightly less effective than the linear progressive schedule**.
>
> | window size | linear progressive | quadratic progressive | random |
> |-------------|--------------------|------------------------|--------|
> | w8          | 82.9 & 2.08       | 82.9 & 2.00           | 82.3 & 2.08 |
> | w16         | 81.7 & 2.17        | 82.9 & 2.04            | 82.3 & 2.13 |
> | w32         | 84.1 & 2.04        | 82.9 & 2.00            | 83.5 & 2.04 |
> | w64         | 83.5 & 1.96        | 82.3 & 1.96            | 83.5 & 2.04 |
>
> ### **W6: The paper does not explicitly discuss limitations or potential failure cases.**
>
> Thank you for the suggestion. We summarize the limitations as follows and will include them in the revised version. First, the acceleration gains may be less pronounced on older-generation GPUs with lower FLOPs. Second, our current implementation does not fully optimize Flex-Attention; during training, we directly rely on FlashAttention, leaving room for further engineering improvements.

---

> > ### Author Response · Authors · 2025-11-28
> >
> > ### **Q1: Could the authors provide deeper intuition for why the linear progressive cyclic noise schedule is particularly effective? Does such noise scheduling methodology possess any conceptual relations to diffusion-style denoising?**
> >
> > Thank you for the insightful question. It is intuitive that a linear progressive noise schedule is more effective than a purely random one: later blocks can condition on earlier blocks that contain fewer noisy tokens, allowing more information to be preserved and providing a stronger and more stable learning signal. This is analogous to diffusion-style training procedures (e.g., Diffusion Forcing for video models [1]), where gradually increasing noise across timesteps or frames leads to more reliable optimization and convergence.
> >
> > As for why the linear schedule outperforms the quadratic one, our findings are primarily empirical. A plausible explanation is that linear progression maintains a more balanced distribution of noise across blocks, whereas quadratic progression may overweight either very low-noise or very high-noise regions, making the training signal slightly harder to optimize.
> >
> > [1] Diffusion Forcing: Next-token Prediction Meets Full-Sequence Diffusion. https://arxiv.org/abs/2407.01392
> >
> > ### **Q2: Beyond the example in Figure 2, can you provide more direct quantitative evidence isolating the contributions of multi-block decoding and rejection recycling to the speedup?**
> >
> > Thanks for the valuable suggestion. We have applied multi-block decoding (MD) and rejection cycling (RC) separately on the math tasks and report the resulting TPF below by fixing the block initialization threshold at $r=0.85$, $K=2$, and scanning over block size = 8, 16, 32, 64, 128, 256. We report the highest TPF achieved. As shown, each component provides a clear gain, and combining both yields the best performance. We will include this ablation in the revised version.
> >
> > |  Strategy | GSM8k | MATH  |
> > |---------|------|----------------|
> > | vanilla Jacobi decoding | 3.62  |  3.66  |
> > | + MD   |  3.71  |  3.74  |
> > | + RC   | 3.88  |  3.86  |
> > | + both   | 4.04 | 3.98   |

---

### Official Review · Reviewer_A7a7 · 2025-11-02

**Soundness:** 3
**Presentation:** 3
**Contribution:** 2
**Rating:** 4
**Confidence:** 4

**Summary:**

The paper proposes pcLLM, a training recipe that turns an autoregressive (AR) LLM into a faster parallel decoder without abandoning causal generation. The key idea is progressive consistency distillation: instead of learning to predict a large block of future tokens at once, the model is trained with a progressive noise schedule and a noise-aware causal mask so that it can reliably predict more correct tokens under noisy unconverged contexts. Building on the resulting trajectory properties, the authors introduce two inference schemes—rejection-recycling of verified n-grams and multi-block decoding—to further increase tokens accepted per iteration. On coding benchmarks (HumanEval, MBPP) with Qwen2.5-Coder-7B-Instruct, pcLLM reports ~3.6–3.8× speedups vs AR with small accuracy deltas, and ~4× with multi-block + rejection-recycling on H200, while outperforming several diffusion-LLM decoders in both speed and pass@1 at the same scale.

**Strengths:**

1. Decouple training a parallel-friendly AR model from inference tricks; relate to Jacobi decoding theory and consistency training.

2. Progressive noise schedule reduces hardest long-noisy contexts; noise-aware causal masks + sequence packing cut passes from O(N) to O(1) for the loss, a practical speedup for training.

3. Lossless inference schemes. Rejection-recycling and multi-block decoding match/verify with greedy acceptance while mining high-quality n-grams that emerge in pcLLM trajectories.

4. Empirical gains (coding). Consistent 3.5–3.8× speedups at similar pass@1; ~4× with multi-block on H200; ablations show block-size effects and noise-schedule importance.

**Weaknesses:**

1. Narrow evaluation. Claims are benchmarked only on HumanEval/MBPP. No natural-language generation, instruction-following, or math/CoT tasks where noisy context tolerance may differ. The introduction strongly motivates broad latency issues (e.g., CoT), but experiments don’t validate there.

2. Baselines could be stronger. While Jacobi and CLLM are included, the paper downplays speculative decoding families (EAGLE/EAGLE-2/3, Medusa, OSD) even though these are the most widely deployed AR accelerators; some are cited but not comprehensively compared in equal settings (same model, context lengths, caching, batch).

3. Systems reporting. Results emphasize TPS and speedup ratios; please detail absolute end-to-end latency (including KV-cache trim/clone, verification kernels, and multi-block overhead) and the FLOPs trade-off vs AR and dLLM baselines. Clarify whether speedups persist under long contexts and with prefix-cache enabled. (Parts are implied, but the precise accounting is unclear.)

14470_AR_Models_can_be_Faster_

Reproducibility gaps: The text claims reproducibility, but training requires Jacobi-trajectory mining, progressive schedules, and custom masking. Without code/configs, there’s residual ambiguity (e.g., window size choices across rounds, exact LR/steps for second pass, acceptance thresholds in Algorithm 1).

14470_AR_Models_can_be_Faster_

Generality claims: The title and framing (“faster and more accurate than diffusion LLMs”) are strong; the dLLM baselines are a subset and mostly at 7B. A balanced claim would qualify the result as “at comparable size on coding benchmarks.”

**Questions:**

1.How is wall-clock latency measured? Please report per-prompt P50/P90 latency including: forward pass, verification, n-gram matching, KV-cache trim/clone, and multi-block spawning/promotion. Also report batch-size sensitivity.

2. Generalization beyond coding. Can you include results on at least one NL generation (e.g., summarization) and one reasoning/CoT benchmark to substantiate broader claims? Your intro motivates CoT latency prominently.

3. Spec-dec baselines. Please add a careful comparison to strong speculative decoding variants (e.g., EAGLE-2/3, Medusa) on the same backbone and hardware, with clear acceptance rates and end-to-end latency. What is pcLLM’s relative FLOPs vs those methods?

4. Ablations on masks/schedules. The linear progressive schedule wins in Table 4 for small-scale ablations; could you show the same with full-data training and larger blocks (e.g., 128/256) to confirm the trend holds?

5. Accuracy trade-offs. On HumanEval, pcLLM (MR) keeps the same pass@1 as pcLLM but both are a bit below AR. Can you discuss when accuracy dips occur (e.g., long dependencies) and whether a small AR fallback on hard blocks recovers them?

---

> ### Author Response · Authors · 2025-11-28
>
> ### **Q1 & W3: How is wall-clock latency measured? Please report per-prompt P50/P90 latency including: forward pass, verification, n-gram matching, KV-cache trim/clone, and multi-block spawning/promotion. Also report batch-size sensitivity.**
>
>
> Latency profiling is done by recording the end-to-end latency for decoding all tokens for each prompt until EOS is reached. Across all 164 prompts from HumanEval on B200 GPU, we provide P50 and P90 latency breakdown on B200 for all operation categories as follows:
>
> | Metric                            | Forward pass | Verification | N-gram + lookahead | KV resize/trim/reindex | Multi-block spawn/promo | Misc (masks, cat/allocs, and others) |
> |-----------------------------------|--------------|--------------|--------------------|------------------------|--------------------------|--------------------------------------|
> | p50 latency (s)                   | 0.643      | 0.028      | 0.014            | 0.021                | 0.019                  | 0.02079                               |
> | p90 latency (s)                   | 0.939      | 0.040      | 0.020            | 0.030                | 0.020                  | 0.030                               |
> | Approx. percentage of total time  | 86.9%        | 3.7%         | 1.9%               | 2.8%                   | 1.9%                     | 2.8%                                  |
>
> We observe that the forward pass accounts for the largest share of runtime, while other operations, including verification, also introduce non-negligible costs. System-level optimizations for KV-cache operations and $n$-gram pool management could further reduce these costs and allow TPF improvements to translate more directly into wall-clock speedups.
>
> We would like to clarify that the profiling result in Figure 4 (a) is batched tokens, therefore reflecting batch-size sensitivity. To further address your concern, we have extended wall-time latency versus batched token profiling on A100 and B200 GPU as well with increasing batch sizes, extending from 16 to 512. You can find further details in Appendix C.
>
> ### **W3: FLOPs trade-off clarification.**
>
> The most important trade-off in pcLLM inference algorithm is TPF vs. FLOPs, i.e. **finding the block size that maximizes TPS on a given hardware budget**. This is because choosing different block sizes determine the degree of parallelism and therefore total FLOPs consumption: larger blocks increase TPF (tokens per forward) by drafting more tokens in parallel but also raise per-step FLOPs.
>
> We perform this analysis and report the resulting TPS curves under different block size choices at 8, 16, 32, 64, 128, and 256. We have elaborated more details on the TPF vs. FLOPs tradeoff in Section 4.3 of our paper.
>
> ### **W1 & Q2: Generalization beyond coding. Can you include results on at least one NL generation (e.g., summarization) and one reasoning/CoT benchmark to substantiate broader claims? Your intro motivates CoT latency prominently.**
>
> Thank you for the suggestion. We have added additional experiments on math tasks, which is natural language generation. Specifically, we initialize the model with Qwen2.5-Math-7B-Instruct, use math problems as the prompt set in OpenThoughts to generate trajectories, and then apply pcLLM training under the same settings as used for the code tasks. The numbers in parentheses indicate the accuracy change compared to the original AR model, and TPF denotes the number of tokens produced per forward pass. Additional throughput and speedup results on A100 GPUs can be found in Table 3 of the updated manuscript.
>
> | Dataset | TPF $\uparrow$  | Accuracy $\uparrow$   |
> |---------|------|----------------|
> | GSM8k   | 4.04 | 91.4 (-1.0)    |
> | MATH    | 3.98 | 77.4 (+0.4)    |

---

> ### Author Response · Authors · 2025-11-28
>
> ### **W2& Q3: Spec-dec baselines. Please add a careful comparison to strong speculative decoding variants (e.g., EAGLE-2/3, Medusa) on the same backbone and hardware, with clear acceptance rates and end-to-end latency. What is pcLLM’s relative FLOPs vs those methods?**
>
> We agree that speculative decoding baselines are highly relevant families of AR acceleration methods, and we have added comparison with two latest SD baseline (EAGLE-3 [1] and HASS [2], which are stronger than Medusa, and Medusa-2).
>
> We also add further comparison with latest dLLM baselines (Fast-dLLM v2 [3], SDAR [4]), as well as the consistency distilled dLLM baseline dParallel (as discussed in response to W1/Q1). **TPS is measured on a single B200 GPU and TPF is hardware independent**.
>
> | Family          | Method        | Acc. $\uparrow$ | TPF $\uparrow$     | TPS $\uparrow$        | Speedup vs. AR $\uparrow$ |
> |-----------------|----------------------|---------------|------------------|--------------------------|-------------------------|
> | AR baseline    | AR (greedy)   | 87.8           | $1.00$           | $83.0$               |   $1.00\times$         |
> | AR + Jacobi        | Jacobi         |   87.8        |  $1.05$         |     $84.7$       |     $1.02\times$        |
> | dLLM            | Fast-dLLM v2    | 63.4          |  $1.00$         |  $83.3$          |    $1.00\times$        |
> | dLLM            | SDAR                | 78.7          |  $2.36$            | $31.5$               | $0.38\times$     |
> | dLLM (distilled) | dParallel       | 54.3          |    $2.90$      |  $175.2$                 | $2.11\times$      |
> | AR + Spec-Dec   | EAGLE-3*    | 68.9*        | $6.38$            | $246.1$          |    $2.97\times$      |
> | AR + Spec-Dec   | HASS*         |  61.6*             | $5.53$       | $280.3$           |   $3.37\times$     |
> | AR + Spec-Dec    | CLLM          | 87.8               | $2.80$        |         $207.4$         | $2.50\times$    |
> | AR + Jacobi         | pcLLM             |     84.8       |     $3.96$     |      $299.5$         |  $3.61\times$       |
> | AR + Jacobi        | pcLLM (MR)     |      84.8      |      $4.21$     |      $\textbf{319.6}$    |       $\mathbf {3.85\times}$  |
>
> For Eagle-3 and HASS, we use the best available checkpoint provided by the authors for speed measurement. For Eagle-3, we use DeepSeek-R1-Distill-Llama-8B as the base model; for HASS, we use LLaMA3-Instruct as the base model.
>
> From the table, we can observe that currently pcLLM achieves both the best TPF as well as a wall-clock speedup at the cost of small performance degradation from the original AR model baseline.
>
> [1] Li, Yuhui, et al. “EAGLE-3: Scaling up Inference Acceleration of Large Language Models via Training-Time Test.” arXiv, 3 Mar. 2025, arxiv.org/abs/2503.01840.
>
> [2] Zhang, Lefan, et al. “Learning Harmonized Representations for Speculative Sampling.” arXiv, 28 Aug. 2024, arxiv.org/abs/2408.15766.
>
> [3] Wu, Chengyue, et al. “Fast-dLLM v2: Efficient Block-Diffusion LLM.” arXiv, 30 Sept. 2025, arxiv.org/abs/2509.26328.
>
> [4] Cheng, Shuang, et al. “SDAR: A Synergistic Diffusion-AutoRegression Paradigm for Scalable Sequence Generation.” arXiv, 7 Oct. 2025, arxiv.org/abs/2510.06303.

---

> ### Author Response · Authors · 2025-11-28
>
> ### **W4: Reproducibility gaps: The text claims reproducibility, but training requires Jacobi-trajectory mining, progressive schedules, and custom masking. Without code/configs, there’s residual ambiguity (e.g., window size choices across rounds, exact LR/steps for second pass, acceptance thresholds in Algorithm 1).**
>
> We thank the reviewer for the question. We clarify data preparation (training sequence packing with noise schedule mapping), training, and inference hyperparameters as follows.
>
> *Progressive Noise Schedule and Training Sequence Packing Details*
>
> To clarify, we map the noise schedule $W$ to a training sequence by splitting the target generation $\mathbf{y}$ into $N$ blocks of size $n$ (as shown in Eq. 4), assigning each block a noise ratio $t_i$ over a window of $w$ blocks (noise schedule has $w$ elements, according to Eq. 7), and then choosing, for every block, the trajectory step whose fraction of unconverged tokens best matches $t_i$. Each noisy block is paired with its corresponding clean block and interleaved in one packed sequence, so a single forward pass (with a custom mask) yields both AR loss on clean blocks and consistency loss on noisy blocks.
>
> We have also added further details as well as an illustration to Appendix B.
>
>
> *Consistency Training Details*
>
> On OpenCodeInstruct, we use 400k trajectories (the same set of trajectories from the first round) collected to launch the additional round of training, but with a **larger block size at 32** (in comparison to using a block size of 16 from round 1). This round of training only starts after the training is complete from the first round with 450k training examples. CLLM uses different training data than ours (CLLM trains the models on GSM8K training set for math, Spider training set for text2SQL, CodeSearchNet for Python coding).
>
> In the second round of training, we choose the combination n=32, w=16 by conducting grid search through possible n=32 combinations. In the rebuttal we provide a more comprehensive grid-search result for the optimal configuration using both block size n=32 and n=64 at varying window sizes as shown in Table R1.
>
>
> | n   | Window | 25k steps (Acc / TPF) | 50k steps (Acc / TPF) | 75k steps (Acc / TPF) | 100k steps (Acc / TPF) |
> |-----|--------|-------------------------|--------------------------|--------------------------|---------------------------|
> | 32  | w32    | 86.0 / 3.17             | 85.4 / 3.42              | 85.4 / 3.46              | 83.5 / 3.62               |
> | 32  | w16    | 86.0 / 3.40             | 86.0 / 3.70              | 85.4 / 3.97              | **84.8 / 4.18**              |
> | 32  | w8     | 84.8 / 3.48             | 84.8 / 3.76              | 83.5 / 4.02              | 82.9 / 4.26               |
> | 64  | w64    | 85.4 / 3.21             | 85.4 / 3.29              | 84.8 / 3.46              | 83.5 / 3.57               |
> | 64  | w32    | 85.4 / 3.51             | 84.8 / 3.77              | 83.5 / 4.18              | **83.5 / 4.44**               |
> | 64  | w16    | 86.0 / 3.47             | 85.4 / 3.83              | 85.4 / 4.08              | 82.9 / 4.52               |
>
> **Table R1: ablation study on training hyperparameter choice for an additional round of progressive consistency distillation. Maximum TPF is measured by applying grid search to inference configurations.**
>
> In Table R1, a few observations can be made:
> - As the block size scales, larger window size becomes a suboptimal choice. And adding noise too aggressively (with the respect to block size) will result in training instability.
> - As training data scales, maximum TPS will saturate among correctly generated samples.
>
> *Inference Hyperparameter Choice details*
>
> We evaluate \sysname (MR) using Jacobi decoding on A100, H200, and B200 GPUs, sweeping block sizes up to $128$ (no latency penalty up to $64$ and only minor degradation at $128$ on H200, as shown from the profiling result in Figure 4a) and fixing accepted tokens to $2,3,4,5$.
>
> For the final inference setup, we use block size $64$, verification size $4$, such that tokens decoded in parallel each step is kept below $256$ tokens, with the initialization threshold fixed at $r=0.85$ (according to more detailed roofline profiling results provided in Appendix C).
>
> We have added a more detailed discussion to Section 4.3 and Appendix C.
>
> You can also find our modeling and profiling scripts in the supplementary materials for reference. Feel free to check it out and we will be more than happy to provide further clarifications.

---

> ### Author Response · Authors · 2025-11-28
>
> ### **W5: Generality claims: The title and framing (“faster and more accurate than diffusion LLMs”) are strong; the dLLM baselines are a subset and mostly at 7B. A balanced claim would qualify the result as “at comparable size on coding benchmarks.”**
>
> Thank you for the comment. We agree that the original title and framing were imprecise. In the revised version, we will qualify our claims to be more balanced, e.g., stating that “pcLLM achieves faster and more accurate results at comparable model sizes.” Also we have added results in response to your Q1 for the generalizability claim.
>
>
>
> ### **Q4: Ablations on masks/schedules. The linear progressive schedule wins in Table 4 for small-scale ablations; could you show the same with full-data training and larger blocks (e.g., 128/256) to confirm the trend holds?**
>
> Thank you for the suggestion. Full-data training is not feasible due to the time limit, but we will include it in the revised version. We conduct additional experiments on the same 10k-prompt set **using a larger block size of 128 and evaluate on HumanEval, and the results are worse compared to using a block size of 16 (Table 4 of the updated manuscript)**.
>
> | Window size | Acc. | Iter/Token |
> |--------|------|-------------|
> | w8     | 84.5 | 0.76        |
> | w16    | 84.3 | 0.76        |
> | w32    | 84.8 | 0.78        |
> | w64    | 84.7 | 0.79        |
>
> ### **Q5: Accuracy trade-offs. On HumanEval, pcLLM (MR) keeps the same pass@1 as pcLLM but both are a bit below AR. Can you discuss when accuracy dips occur (e.g., long dependencies) and whether a small AR fallback on hard blocks recovers them?**
>
> Thank you for this interesting feedback. We would like to clarify that our Jacobi decoding algorithm, with multi-block decoding and rejection sampling, **provably recovers the same outputs as AR decoding**, so incorporating an AR fallback would not provide additional benefit. The slight accuracy degradation observed in some settings stems from training-related variance rather than from the decoding procedure itself. Moreover, our additional experiments on math reasoning tasks (results provided in our response to your question Q1) show that this variance remains **small and within an acceptable range**.

---

### Author Response · Authors · 2025-12-03

Dear AC and all reviewers, we appreciate the time and effort you have invested in carefully reading our submission and providing constructive feedback. Below we summarize (1) the key strengths you highlighted, (2) the clarifications we have made in response, and (3) the additional experiments we have conducted to address reviewers questions.

## Strengths acknowledged

- **Conceptual novelty in training: progressive noise schedule, masking, packing (A7a7, TYFF, hCKy):** The progressive noise schedule, noise-aware causal mask, and sequence packing were acknowledge as thoughtful design choices that directly tackle training under "noisy" unconverged contexts and make the training pipeline more stable and efficient.

- **Conceptual novelty in Inference algorithm: rejection-recycling and multi-block decoding (A7a7, XcmS):** The proposed rejection-recycling and multi-block decoding were recognized as principled, (effectively) lossless ways to reuse high-quality n-grams and increase accepted tokens per iteration while remaining compatible with standard greedy decoding.

- **Empirical speedups (A7a7, XcmS, TYFF, hCKy):** Reviewers acknowledged that pcLLM achieves up to $3.8\times$ wall-clock speedup on coding benchmarks (HumanEval, MBPP) with modest accuracy changes, and that multi-block decoding reaches close to $4\times$ speedup, with pcLLM (MR) being both faster and more accurate than CLLM on MBPP.

- **Presentation and structure (XcmS, hCKy):** The paper was described as generally well-structured and clearly motivated, with an intuitive progression from preliminaries to methodology, with effective figures and ablations.

---
## Clarifications made

- **Conceptual relationship with dLLMs (Reviewer XcmS, TYFF):**
  - We have added more extensive discussion on discrete diffusion models and their relation with our work.
  - We provided deeper intuition for why the linear progressive cyclic noise schedule is effective and discussed its conceptual relationship to diffusion-style denoising.
  - We clarified multi-block decoding algorithm, and why starting future blocks before earlier ones are fully denoised can still lead to speedup (since they generate draft tokens that can be verified later and lead to fast forwarding).

- **Mapping noise schedule to training sequence (A7a7, TYFF):**
  - We clarified how the noise ratios $t_i$ are used in practice and how blocks are sampled from Jacobi trajectories and combined so that each training block approximately matches the desired noise schedule.
  - We detailed how additional Jacobi trajectories are collected, how much data used on each stage of training, and how block sizes are progressively increased during training (along with ablation study to justify our choice).

- **Hyperparameter choices how they affect speedup and accuracy (Reviewer A7a7, XcmS, TYFF, hCKy):**
  - We precisely defined window size $w$, block size $n$, and their relationship, and clarified the choice of block sizes used for the main results table (as mentioned in the original paper).
  - We discussed sensitivity to inference hyperparameters (spawn ratio $r$, number of maximum active blocks $K$, block size $n$, and verification pool size).
  - We clarified the TPF-FLOPs trade-off across different block sizes and how different choices of block size affect end-to-end latency (larger blocks lead to higher TPF but also require more FLOPs).

- **Wording (Reviewer A7a7, TYFF, hCKy):**
  - We softened and made more precise the claim of being “faster and more accurate than diffusion LLMs,” explicitly restricting it to comparable-size models on coding (and math based on additional experiments) benchmarks and clarifying the scope of our generality claims.
  - We improved the discussion of limitations, reproducibility (training configurations, implementation details), and how the method is expected to scale to larger models.

---
## Additional experiment results presented

- **Broader task coverage (Reviewer A7a7, XcmS, hCKy)**
  - Evaluated on math tasks with longer generations (up to 2048 tokens) to test generalizability.

- **Baseline expansion (A7a7, XcmS, TYFF, hCKy)**
  - Included strong speculative decoding and multi-token prediction baselines (e.g., EAGLE-3, HASS) reporting acceptance length, end-to-end latency as well as generation quality.
  - Compared against distilled diffusion LLMs, dParallel, and more recent dLLM variants, fast-dLLM v2 and SDAR.

- **More comprehensive ablations and inference configuration discussion (A7a7, XcmS, TYFF, hCKy)**
    - More extensive ablations of noise schedules for training (beyond reverse schedule) including more quadratic progressive and random schedules.
    - Added comprehensive sweeping over block size $n$ (8, 16, 32, 64, 128, 256) and plot block size (FLOPs) vs. speedup (TPS) Pareto curves for both pcLLM and CLLM.
    - We clarified how wall-clock latency is measured and latency breakdown (including forward passes, verification, KV-cache operations, and others).

---

### Meta-Review · Area_Chair_DDWr · 2026-01-07

**Summary:**

This paper proposes pcLLM, a progressive consistency distillation framework that adapts autoregressive models into parallel decoders, along with inference-time mechanisms such as rejection recycling and multi-block decoding. Reviewers agreed that the paper addresses an important problem, reducing inference latency, and contains several technically interesting ideas. However, the initial reviews raised substantial concerns regarding clarity, evaluation completeness, baseline selection, and the interpretation of speed–accuracy trade-offs.

**Reviewer Concerns:**

Reviewers raised concerns about the limited initial evaluation scope (primarily coding benchmarks), missing or late inclusion of strong speculative decoding baselines, and unclear reporting of absolute wall-clock latency and FLOPs trade-offs. Several reviewers found the method difficult to follow and questioned reproducibility due to the complexity of the training pipeline. The rebuttal substantially expanded the experimental section, but did not fully address concerns about the added complexity and retraining cost relative to the observed gains, or that the conclusions generalize beyond the specific settings evaluated.

**Reviewer Scores:**

Reviewer scores initially spanned from strong reject to marginally below the acceptance threshold, reflecting divergent views on clarity, novelty, and practical impact. Overall, despite the rebuttal, the score distribution is likely to remain mixed and does not clearly support acceptance.

---

### Decision · Program_Chairs · 2026-01-26

Reject